# Phytoplankton reaction to an intense storm in the north-western Mediterranean Sea

Stéphanie Barrillon[1], Robin Fuchs[1,2], Anne A. Petrenko[1], Caroline Comby[1], Anthony Bosse[1], Christophe Yohia[3], Jean-Luc Fuda[1], Nagib Bhairy[1], Frédéric Cyr[4], Andrea M. Doglioli[1], Gérald Grégori[1], Roxane Tzortzis[1], Francesco d'Ovidio[5], and Melilotus Thyssen[1]

[1]Aix Marseille Univ., Université de Toulon, CNRS, IRD, MIO UM 110 , 13288, Marseille, France
[2]Aix Marseille Univ, CNRS, Centrale Marseille, I2M, Marseille, France
[3]Aix Marseille Univ., Université de Toulon, CNRS, IRD, OSU Pytheas UAR 3470 , 13288, Marseille, France
[4]Fisheries and Oceans Canada, Northwest Atlantic Fisheries Centre, St. John's, Canada
[5]LOCEAN, UMR CNRS / Université P. et M. Curie / IRD / MNHM, F-75005 Paris, France

**Correspondence:** Stéphanie Barrillon (stephanie.barrillon@mio.osupytheas.fr)

**Abstract.** The study of extreme weather events and their impact on ocean physics and biogeochemistry is challenging due to the difficulty of collecting in situ data. Yet, recent research pointed out the major influence of such physical forcing events on microbiological organisms. Moreover, such intense event occurrences may rise in the future in the context of global change. In May 2019, an intense storm occurred in the Ligurian Sea (north-western Mediterranean Sea) and was captured during
the FUMSECK cruise. In situ multi-platform measurements (vessel-mounted ADCP, thermo-salinometer, fluorometer, flow cytometer, a Moving Vessel Profiler equipped with a multi-sensor towed vehicle, and a glider) along with satellite data and a 3D atmospherical model were used to characterise the fine-scale dynamics occurring in the impacted oceanic zone. The most affected area was marked by a lower water temperature (1°C colder), and an increase by a factor of two in surface chlorophyll-a and of seven in nitrate concentrations, exhibiting strong gradients with respect to the surrounding waters. Our results show that
this storm led to a deepening of the mixed layer depth from 15 to $50\,\mathrm{m}$ and a dilution of the deep chlorophyll maximum. As a result, the surface biomass of most phytoplankton groups identified by automated flow cytometry increased by up to a factor of two. Conversely, the carbon-chlorophyll ratio of most phytoplankton groups dropped down by a factor of two, evidencing significant changes in the phytoplankton cell composition. These results suggest that the role of storms on the biogeochemistry and ecology of the Mediterranean Sea may be underestimated and highlight the need, during these events, for high-resolution
measurements coupling physics and biology.

# 1 Introduction

Marine environments are subject to short-term events whose effects on biogeochemical processes can be substantial. This is the case of desert dust deposit on oligotrophic areas (Guieu et al., 2014), volcanic ash deposit (Hamme et al., 2010), submarines sources of iron (Guieu et al., 2018), and sudden mixing of the water column from typhoons (Wang, 2020). Even the classical phytoplankton spring bloom can vary in intensity and spatial extent depending on the amount of previous short-term storms (Ferreira et al., 2022). The effect of these processes on marine micro-organisms such as phytoplankton includes sudden changes in diversity and abundance. Depending on the redistribution of nutrients, the turbulence, the light conditions and the mixing of different water masses, the phytoplankton community can collapse or grow, affecting carbon export by generating decoupling phenomena between production and remineralisation (Henson et al., 2019).

Meteorological impulse wind events, such as storms, and their effects on oceanic physics and on biogeochemistry, are poorly explored with in situ data. Such events generate mixing and stirring of the surface layer and can trigger transitional peaks in primary production, mainly explained by nitracline shoaling and grazers dilution (Lomas et al., 2009; Menkes et al., 2016). In oligotrophic ocean conditions, Babin et al. (2004) and Han et al. (2012) observed sudden and large increases in chlorophyll-a (chla) from satellite ocean color, lasting several weeks after summer hurricane-storms. The resulting increase in chla integrated over the first few meters reached values close to those from the spring bloom (Babin et al., 2004) with potential primary production comparable to the one induced by some mesoscale ($\sim 10 - 100\,\mathrm{km}$ horizontal range) eddies. Nevertheless, the authors were limited in their interpretation by the lack of in situ observations. Only a few studies have combined high-resolution physical descriptions of wind events with a phytoplankton resolution at the functional group level. Some coastal studies, such as Fuchs et al. (2022), have evidenced pico-nanophytoplankton abundance and biomass responses, positive for most phytoplankton groups, within two to four days following wind-induced events at a coastal station located in the north-western (NW) Mediterranean Sea in stratified conditions. The authors showed that extreme events can generate daily biomass increases of the same order of magnitude as those observed during the spring bloom. Similarly, Anglès et al. (2015) studied the response of nano-microphytoplankton to tropical cyclones generating wind-physical forcing and substantial rains in the Western Gulf of Mexico. They highlighted strong increases in plankton abundance following the storms with delays consistent with Fuchs et al. (2022). These storms observed in either coastal Mediterranean systems or tropical open ocean may also exert a strong control on both primary production and community structure in the Mediterranean open ocean, thus playing a potentially important biogeochemical role on the whole basin. However, to our knowledge, no such event in the Mediterranean open ocean has been reported in the past.

The classical spring bloom as observed in temperate oceans is triggered by the shoaling of the mixed layer when passing from the winter convection to the spring stratification (Behrenfeld, 2010). The bloom ends when no more nutrients are available in the euphotic layer or when grazers overpass the phytoplankton growth capacity. This is particularly the case in the NW Mediterranean Sea characterised by winter deep convection (Houpert et al., 2016; Testor et al., 2018) and by spring blooms

of different intensities that can be detected from satellite images (d'Ortenzio and Ribera d'Alcalà, 2009; Mayot et al., 2016). This area is affected by strong northerly winds, which strength in winter defines the bloom intensity (Conan et al., 2018). In summer stratified conditions, impulse wind events could induce submesoscale ($\sim 1 - 10\,\mathrm{km}$ horizontal range) vertical mixing and trigger patches of high phytoplankton production. Yet, observing the effect of these events on phytoplankton dynamics and distribution is challenging, especially during stratified oligotrophic conditions, and requires the deployment of dedicated automated and high-frequency sampling tools. Indeed, the mixing of the water column may bring microorganisms from deep to surface layers, affect their physiological properties due to photoacclimation processes, and impact their carbon-to-chlorophyll ratios used to run primary production models at large scales (Sathyendranath et al., 2020). In addition, some scarce observations at the functional group level evidence daily adaptation processes rather than community changes after water column mixing (Thompson et al., 2018) or a taxonomical dependency in physiological strategies (Graff and Behrenfeld, 2018). Being able to monitor phytoplankton distribution at a functional level, by integrating small and rapid scale dynamics into larger space and time, would precise the role of phytoplankton in biogeochemical processes.

The objective of this paper is to study in situ physical and biological effects of a particularly intense episode of wind that occurred in spring 2019 in the Ligurian Sea (NW Mediterranean Sea) during the FUMSECK cruise (Facilities for Updating the Mediterranean Submesoscale - Ecosystem Coupling Knowledge, https://doi.org/10.17600/18001155, Barrillon et al. (2020)). High-resolution physical properties in the water column and surface phytoplankton functional groups distribution were combined to show abrupt changes in water characteristics, surface phytoplankton abundances and physiological indicators.

## 2   Material and methods

The FUMSECK cruise aimed at simultaneously sampling physical and biogeochemical data for the study of mesoscale and submesoscale dynamics, which imply structures such as eddies, filaments, or fronts over a horizontal spatial range of 1 to $100\,\mathrm{km}$, a vertical one of 0.1 to $1\,\mathrm{km}$, and a temporal range of days to a few weeks (Giordani et al., 2006; Ferrari and Wunsch, 2009; McWilliams, 2019). It took place from 30 April 2019 to 7 May 2019, in the Ligurian Sea (NW Mediterranean Sea), onboard the RV Téthys II. The circulation in the Ligurian Sea is generally cyclonic and characterised by a strong westward flowing geostrophic current along the coastline (Esposito and Manzella, 1982). The Northern Current (Millot, 1999), hereafter called NC, corresponds to the Northern branch of the current along the coastline. During this cruise, we deployed towed instruments (Moving Vessel Profiler, MVP) and an underwater glider (Testor et al., 2019) to measure physical properties at high resolution. These measurements have been paired with shipboard measurements of phytoplankton functional groups from an automated pulse-shape recording flow cytometer, based on cell sizes and pigment contents (Dugenne et al., 2014; Thyssen et al., 2014; Bonato et al., 2015; Louchart et al., 2020). Figure 1 shows the cruise and the glider trajectories, the MVP transects and the positions of the surface discrete sampling for nutrients and chla. A storm hit the region between 4 and 5 May. Right after the storm, during which we had to take shelter, the ship came back to the wind-exposed zone to collect data. Meanwhile,

the glider stayed in the storm-exposed zone all along and collected data. In addition to in situ data, we exploited satellite data to guide the cruise and obtain a synoptic view of the region and a meteorological model to study the storm.

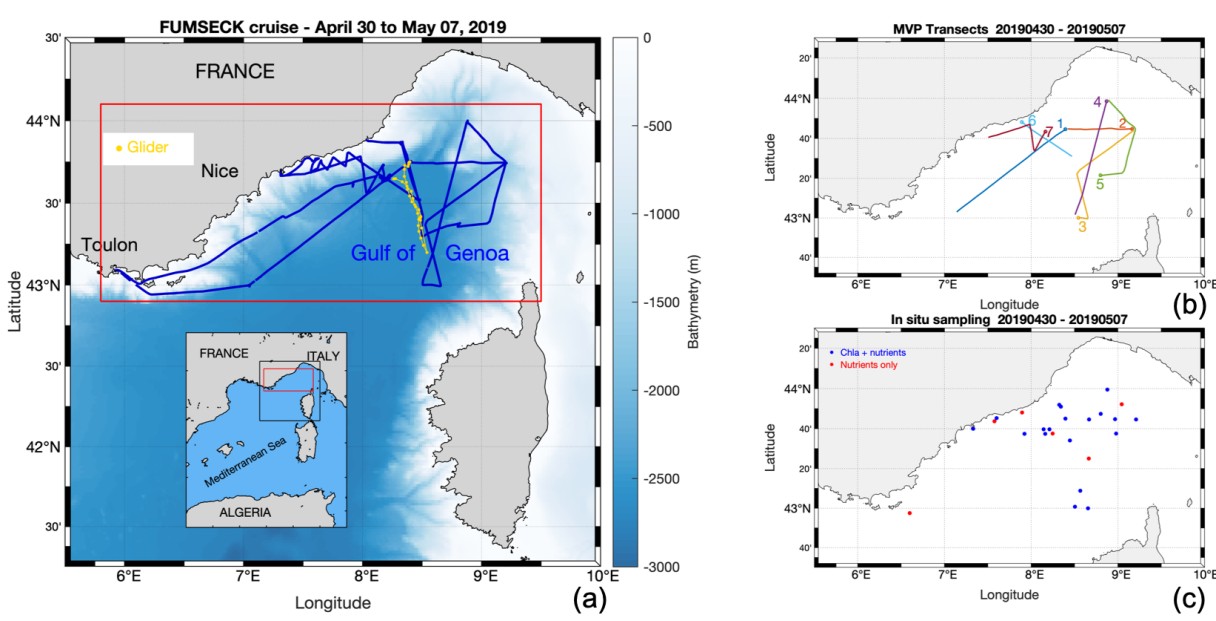

**Figure 1.** (a) FUMSECK cruise (blue line), superimposed with the bathymetry. The geographical domain is represented in red and the glider trajectory in yellow. (b) MVP transects, numbered from 1 to 7 at the end of the transects. (c) In situ sampling of chla and nutrients.

## 2.1 Transect measurements

The vessel-mounted Acoustic Doppler Current Profiler (VM-ADCP, RDI Ocean Surveyor 75 kHz) continuously acquired data during the cruise. The vertical range was set to $[18\,\mathrm{m};\,562\,\mathrm{m}]$ with a $8\,\mathrm{m}$ resolution. Current data were averaged and stored every 2 minutes, corresponding to a horizontal resolution of $0.4\,\mathrm{km}$ for a vessel speed of 6.3 knots. The resulting horizontal currents have been processed with Cascade 7.2 software (Le Bot et al., 2011).

The surface-water flow-through system pumped seawater at a $2\,\mathrm{m}$ depth with a flow rate of about $60\,\mathrm{L\,min^{-1}}$. A thermosalinograph (TSG, SeaBird SBE 21) acquired sea surface temperature (conservative, $\Theta$\_tsg) and salinity (absolute, S\_tsg) data every minute. A fluorometer (Turner Designs, 10-AU-005-CE) recorded simultaneously sea surface red fluorescence $> 680\,\mathrm{nm}$ after excitation in the blue (Rfluo\_tsg (a.u.), a.u. standing for arbitrary units) as a proxy of chla content.

The MVP200 was deployed with the Multi Sensor Free Fall Fish (MSFF) set of instruments, including a $\mu$CTD (AML S/n 7373 PDC-B0204). Temperature (conservative, $\Theta$\_mvp) and salinity (absolute, S\_mvp) profiles were treated with the La-

texTool Package (Doglioli and Rousselet, 2013). In total, 507 profiles were performed over $680.4\,\mathrm{km}$ of route (58h25min of effective measurements), separated in 7 transects (MVP 1 to 7) with a mean duration of 8h20min each and a mean vessel speed of 6.3 knots.

Along the cruise, 26 samples for surface Phosphate ($PO_4^{3-}$), Nitrate ($NO_3^-$), Nitrite ($NO_2^-$) and Silicate ($Si(OH)_4$) concentrations were collected from the flow-through system in $20\,\mathrm{mL}$ high-density polyethylene bottles poisoned with HgCl2 to a final concentration of $20\,\mathrm{mg\,L^{-1}}$ and stored at $4°C$ before being analysed in the laboratory a few weeks later. Nutrient concentrations were determined using a Seal AA3 auto-analyser following the method of Aminot and Kérouel (2007) with

105 analytical precision of $0.01\,\mu mol\,L^{-1}$ and quantification limits of 0.02, 0.05 and $0.30\,\mu mol\,L^{-1}$ for $PO_4^{3-}$, $NO_3^-$ (and $NO_2^-$) and $Si(OH)_4$, respectively.

Similarly, surface chla concentration (Chl_insitu, $\mathrm{ng\,mL^{-1}}$) was extracted from a total of 20 samples filtered from $500 \pm 20\,\mathrm{mL}$ of seawater through $25\,\mathrm{mm}$ glass-fiber pyrolysed filters (Whatman® GF/F) and immediately frozen at $-20°C$. Filters were

110 placed in glass tubes containing $5\,\mathrm{mL}$ of pure methanol and allowed to extract for 30 min as described by Aminot and Kérouel (2007). The fluorescence of the extract was determined by using a Turner Fluorometer AU10 equipped with the Welschmeyer kit to avoid chlorophyll-b interference (Welschmeyer, 1994). The fluorometer was zeroed with a methanol turbidity blank. The detection limit was $0.01\,\mathrm{ng\,mL^{-1}}$. Calibration was performed using a pure chla standard (Sigma Aldrich®, ref: C5753, pure spinach chlorophyll).

Phytoplankton abundances and functional groups were resolved using an automated pulse-shape recording flow cytometer, a Cytosense (AFCM, cytobuoy b.v.; NL) connected to the flow-through system, which automatically analysed samples for phytoplankton counts in the size range of $0.6-800\,\mu m$ in width. The cells contained in a volume of water were first surrounded by an isotonic sheath fluid, aligned in a laminar flow, and went through a $488\,\mathrm{nm}$ laser beam thanks to a weight-calibrated

sample peristaltic pump. Doing so, a set of optical curves, called pulse shapes, was generated for each cell. The pulse shapes of side-ward scatter (SWS, $488\,\mathrm{nm}$) and fluorescence emissions were separated by a set of optical filters (orange fluorescence (FLO, 552–652 nm) and red fluorescence (FLR, $> 652\,\mathrm{nm}$) and collected on photomultiplier tubes. The pulse shapes of forward scatter (FWS) were collected on left and right-angle photodiodes and used to validate the laser alignment. A total of 400 samples were acquired with a 20-minute time resolution, corresponding to a mean resolution of $3.9\,\mathrm{km}$ during the transects.

The samples were stabilised in a $300\,\mathrm{mL}$ sub-sampling chamber before the acquisition. The instrument and the acquisition protocol are described in Marrec et al. (2018).

For the identification of phytoplankton groups, two protocols were successively run, one triggering on FLR $6\,\mathrm{mV}$ for 5 min targeting Orgpicopro and a second one triggering on FLR25 for 10 min targeting the Redpicoeuk, Rednano, Orgnano, and

130 Redmicro phytoplankton groups (Appendix A). Phytoplankton groups were manually classified using the CytoClus® software by generating several two-dimensional cytograms plotting descriptors of the four pulse shapes such as the area under the curve

of the pulse-shape signals (FWS_cyto, SWS_cyto, Ofluo_cyto, Rfluo_cyto). Group abundances and cell properties were processed by the software.

The size of the different phytoplankton cells was estimated based on the relationship between silica beads real sizes (1.0, 2.01, 3.13, 5.02, 7.27 $\mu$m non-functionalised silica microspheres, Bangs Laboratories, Inc.) and FWS_cyto signal and converted into equivalent spherical diameter (ESD) and biovolume (BV, $\mu$m$^3$). A power-law relationship ($\log(\text{BV}) = 0.912 \times \log(\text{FWS\_cyto})$ $-5.540$, r$^2 = 0.89$, n = 7) allowed the conversion of the FWS signal into cell size. The stability of the optical unit and the flow rates were checked using Beckman Coulter Flowcheck™ fluorospheres (2 $\mu$m) before, during, and after installation. Phyto-
plankton biomass per group were computed in pgC mL$^{-1}$ from the power law of the form aBV$^{\text{b}}$, to get a mean carbon cellular quota (C, pgC cell$^{-1}$), with a and b conversion factors reported by Menden-Deuer and Lessard (2000) and Verity et al. (1992).

## 2.2    Glider

An autonomous Alseamar's SeaExplorer glider was deployed during the whole cruise in order to perform complementary measurements on the dynamics and biogeochemistry around the area of the cruise. It performed saw-tooth cycles with a pitch angle
of about $20 - 25°$ from the surface to $600$ m depth in about $2$ h, resulting in a distance between consecutive vertical profiles of about $1$ km. The glider was equipped with a pumped Seabird CTD probe (Glider Payload CTD), and a Wetlab ECO-puck with chla fluorescence channel sampling at $0.25$ Hz, corresponding to a vertical resolution of $0.5 - 0.8$ m.

The raw counts from the ECO-puck were converted to chla fluorescence using the manufacturer's calibration coefficients and
were then corrected near the surface during daylight time from non-photochemical quenching following (Xing et al., 2012). To do so, the mixed layer depth was evaluated using a $0.1°$C criterion on the conservative temperature profiles relative to a reference depth of $10$ m (Houpert et al., 2015). The relative differences in fluorescence are used as a quantitative proxy of the evolution in the distribution of the chla concentration. The glider fluorescence data have not been calibrated against reference measurements, but agree well with the surface measurements of the ship's adjusted chla concentrations (Appendix B).

## 155    2.3    Satellite data

The FUMSECK cruise benefited before, during, and after the cruise from the automatic SPASSO software (https://spasso.mio.o supytheas.fr, last access 9 June 2022), which performs real-time processing of CMEMS satellite products (Nencioli et al., 2011; d'Ovidio et al., 2015; Petrenko et al., 2017). The onshore team interpreted the results and sent their daily recommendations on the routes to be taken and the choice of stations to target specific oceanic fine-scale processes like fronts or eddies (Doglioli
et al., 2013; Petrenko et al., 2017). Near-real-time products of Sea Surface Height (SSH) and associated geostrophic currents, Sea Surface Temperature (SST), and chla concentration, together with Lagrangian calculations such as FSLE (Finite-Size Lyapunov Exponents) have been used daily from 2 April 2019 to 3 July 2019, and all the results are available online on https://spasso.mio.osupytheas.fr/FUMSECK/. A total of 11 daily bulletins (from 23 April to 7 May) have been released

and are available online on https://spasso.mio.osupytheas.fr/FUMSECK/Bulletin_web/. The satellite products exploited for FUMSECK are detailed in Appendix C.

## 2.4 Meteorological model

The WRF (Weather Research and Forecasting) model, a non-hydrostatic model developed by NCAR (Skamarock et al., 2019), was run with the core ARW (Advanced Research Weather). The horizontal resolution was $2\,\mathrm{km}$, and the vertical grid was defined with 34 vertical levels. The ARAKAWA-C grid was used one-way with 350 points in the zonal direction and 280 points in the meridional direction. ARW was forced every six hours by the ECMWF (European Centre for Medium-Range Weather Forecasts) coupling model (Bechtold et al., 2020; Ben-Bouallegue, 2020).

The surface net heat flux and winds were extracted from the model at hourly outputs to characterise the storm event. The net heat flux from the atmosphere to the land/sea surface was computed as : $Q_{net} = Q_{sw} + Q_{lw} + Q_{sens} + Q_{lat}$ with respectively $Q_{sw}$, $Q_{lw}$ the short-wave and long-wave radiations, $Q_{sens}$ the sensible and $Q_{lat}$ the latent heat flux. All fluxes are here downward positive.

## 2.5 Fluorescences and chlorophyll-a

Different techniques to estimate chla concentration were used during the cruise and compared. Absolute chla concentration from Chl_insitu was used as the reference to convert red fluorescence from AFCM and TSG fluorometer into chla concentration based on the significant correlations between them (Fig. 2a).

Fluorescence from the TSG (RFluo_tsg) was converted into units of chla concentration (Chl_tsg, $\mathrm{ng\,mL^{-1}}$) using the significant correlation with Chl_insitu, $\mathrm{Chl\_insitu} = 0.85 \times \mathrm{Rfluo\_tsg} - 0.19$, $r^2 = 0.79$, $n = 20$. AFCM chla concentration (Chl_cyto) was estimated from the Rfluo_cyto. Values were normalised with $2\,\mu\mathrm{m}$ Polyscience beads, and multiplied by the abundance of each group to get the total normalised Rfluo_cyto per unit of volume (nRFluo_cyto $(\mathrm{a.u\,mL^{-1}})$). nRFluo_cyto was then compared to the Chl_insitu (Fig. 2a and b). A set of samples from a minicosm experiment (PIANO, unpublished data), acquired with the same chla extraction protocol and the same Cytosense instrument was added to the observations. These samples presented higher chla concentration values, strengthening the relationship. The linear relation between nR-fluo_cyto and Chl_insitu was used to estimate chla concentration for each AFCM phytoplankton group (Chl_cyto, $\mathrm{ng\,mL^{-1}}$), $\mathrm{Chl\_insitu} = 0.11 \times \mathrm{nRFluo\_Cyto}$, $r^2 = 0.97$, $n = 41$ (Fig. 2b). The origin of the linear regression was not significantly different from zero.

Chla concentration from three different satellite ocean color algorithms (Chl_ACRI, Chl_MEDOCL3, Chl_MEDOCL4) were compared to the other sets of chla concentration estimates for sea surface chla validation (Fig. 2a). We performed the association between each Chl_tsg data and the corresponding Chl satellite data on the same day and for the closest lat/lon pixel, then selected the ones where the Chl_tsg data are between 6:00-18:00 UTC daytime, to minimise the effect of night

extrapolated points. The glider sampling did not follow the ship's route, but a comparison of the $0 - 5\,\mathrm{m}$ signal when the ship-to-glider distance was smaller than $15\,\mathrm{km}$ showed a negligible difference with the ship-adjusted surface chla concentrations $(0.04 \pm 0.13\,\mathrm{ng\,mL^{-1}})$.

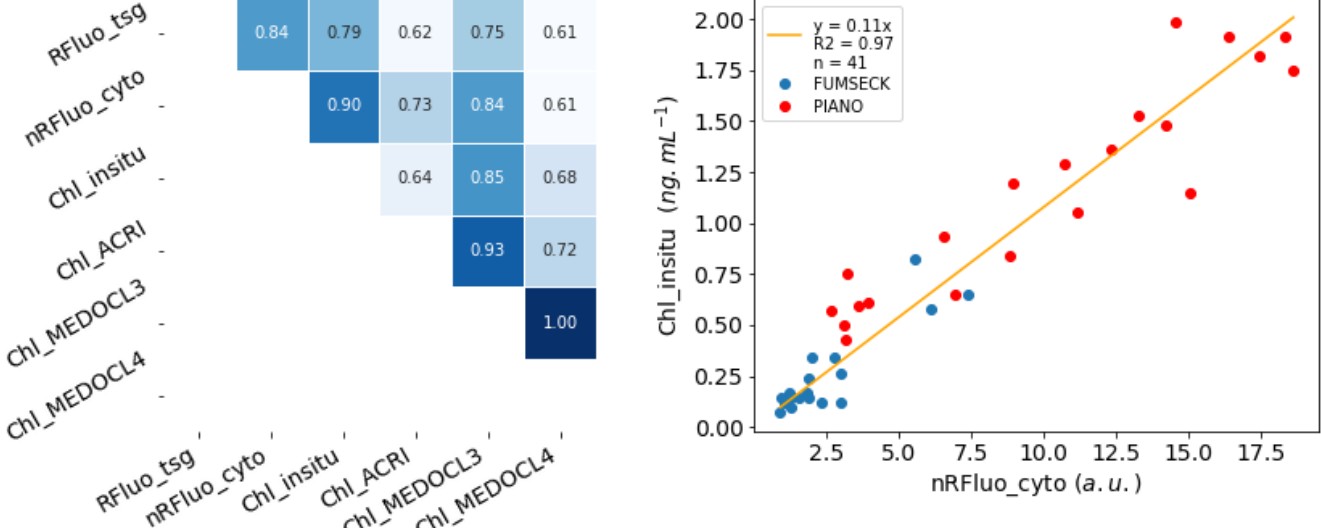

**Figure 2.** (a) Correlation plot between different sources of fluorescence and chla concentration estimation per unit of volume: fluorescence from the flow-through fluorometer (Rfluo_tsg (a.u.), n $= 8385$), sum of all phytoplankton cells normalised red fluorescence from the Cytosense (nRFluo_cyto (a.u mL$^{-1}$), n $= 400$), chla from in situ discrete sampling (Chl_insitu (ng mL$^{-1}$), n $= 20$), from the ACRI ocean color product for the 6:00-18:00 UTC day time (Chl_ACRI (ng mL$^{-1}$), n $= 3522$), from the MEDOCL3 product for the 6:00-18:00 UTC day time (Chl_MEDOCL3 (ng mL$^{-1}$), n $= 2094$), and from the MEDOCL4 product for the 6:00-18:00 UTC day time (Chl_MEDOCL4 (ng mL$^{-1}$), n $= 4498$). All the presented correlations were significant at a 0.01 level using a Pearson test. (b) Linear regression between the chla concentration from in situ discrete sampling (Chl_insitu (ng mL$^{-1}$), n $= 41$) and the sum of all phytoplankton cells normalised red fluorescence from the Cytosense (nRFluo_cyto (a.u mL$^{-1}$)). Two data sets are shown using the same instrument (PIANO and FUMSECK). The intercept coefficient of the regression was not significant at 10% level (t-test).

All the measurements described above are summarised in Table 1.

**Table 1.** Summary of the variables measured during the cruise, including their sources, their sampling spatial and temporal resolution, and the vertical range along which they were measured.

| Observable | Abbreviation / name | Vertical Range | Sampling | Source |
|---|---|---|---|---|
| Horizontal currents | ADCP currents | $18 - 562\,\text{m}$ | all cruise, $0.4\,\text{km}$ resolution | VM-ADCP |
| | geostrophic currents | first meters | daily, 2 April to 3 July 2019 | Satellite |
| Conservative Temperature | $\Theta\_\text{tsg}$ | $2\,\text{m}$ | all cruise, $0.2\,\text{km}$ resolution | TSG |
| | $\Theta\_\text{mvp}$ | $0 - 308\,\text{m}$ | 7 transects, $1.3\,\text{km}$ resolution | MVP |
| | $\Theta\_\text{glider}$ | $0 - 600\,\text{m}$ | 2 transects, $1\,\text{km}$ resolution | Glider |
| Absolute Salinity | $S\_\text{tsg}$ | $2\,\text{m}$ | all cruise, $0.2\,\text{km}$ resolution | TSG |
| | $S\_\text{mvp}$ | $0 - 308\,\text{m}$ | 7 transects, $1.3\,\text{km}$ resolution | MVP |
| | $S\_\text{glider}$ | $0 - 600\,\text{m}$ | 2 transects, $1\,\text{km}$ resolution | Glider |
| Fluorescence | RFluo_tsg | $2\,\text{m}$ | all cruise, $0.2\,\text{km}$ resolution | TSG |
| | RFluo_cyto | $2\,\text{m}$ | 400 samples, $3.9\,\text{km}$ resolution | AFCM |
| | $FL_\text{npq}$ | $0 - 600\,\text{m}$ | 2 transects, $1\,\text{km}$ resolution | Glider |
| Chlorophyll-a | Chl_tsg (converted) | $2\,\text{m}$ | all cruise, $0.2\,\text{km}$ resolution | TSG |
| | Chl_insitu | $2\,\text{m}$ | 20 samples | in situ |
| | Chl_cyto (converted) | $2\,\text{m}$ | 400 samples, $3.9\,\text{km}$ resolution | AFCM |
| | Chl_ACRI | | | |
| | Chl_MEDOCL3 | first meters | daily, 2 April to 3 July 2019 | Satellite |
| | Chl_MEDOCL4 | | | |
| Nutrients | Phosphate ($PO_4^{3-}$) | | | |
| | Nitrate ($NO_3^-$) | $2\,\text{m}$ | 26 samples | in situ |
| | Nitrite ($NO_2^-$) | | | |
| | Silicate ($Si(OH)_4$) | | | |
| Phytoplankton | Abondance, size, biovolume, biomass | $2\,\text{m}$ | 400 samples, $3.9\,\text{km}$ resolution | AFCM |
| Wind intensity | $U_{10}$ | 10 m above surface | all cruise, hourly, $2\,\text{km}$ resolution | model |
| Heat Flux | $Q_\text{net}$ | surface | | |

# 3 Results

## 3.1 Overall circulation

The general oceanic circulation during the FUMSECK cruise is schematised in Fig. 3. In Fig. 3a the horizontal current velocities averaged over $25 - 150$ m are shown for the 7 stations at which the ship stopped, superimposed with the mean chla concentration measured by satellite (Chl_MEDOCL4) from 1 to 6 May 2019. Horizontal current velocities were obtained with the vessel-mounted ADCP, averaged during the 20 min preceding the arrival at each station. The boundaries of the different hydrodynamic zones were drawn based on Chl_MEDOCL4 concentration isolines. The region of the NC (hatched in purple, $< 0.12 \, \text{ng mL}^{-1}$) corresponds to the lowest Chl_MEDOCL4 concentration. The southeastern part of the cyclonic recirculation (hatched in orange, $> 0.15 \, \text{ng mL}^{-1}$) shows the highest Chl_MEDOCL4 concentrations. These two zones are separated by a region, hereafter referred to as the intermediate zone (hatched in green, $0.1 - 0.15 \, \text{ng mL}^{-1}$). The vessel-mounted ADCP horizontal currents at $26.5$ m-depth along the cruise (Fig. 3b) show the high-intensity of the NC ($0.43 \, \text{m s}^{-1}$ mean velocity in the core of the NC) with respect to the cyclonic recirculation zone ($0.18 \, \text{m s}^{-1}$ mean velocity).

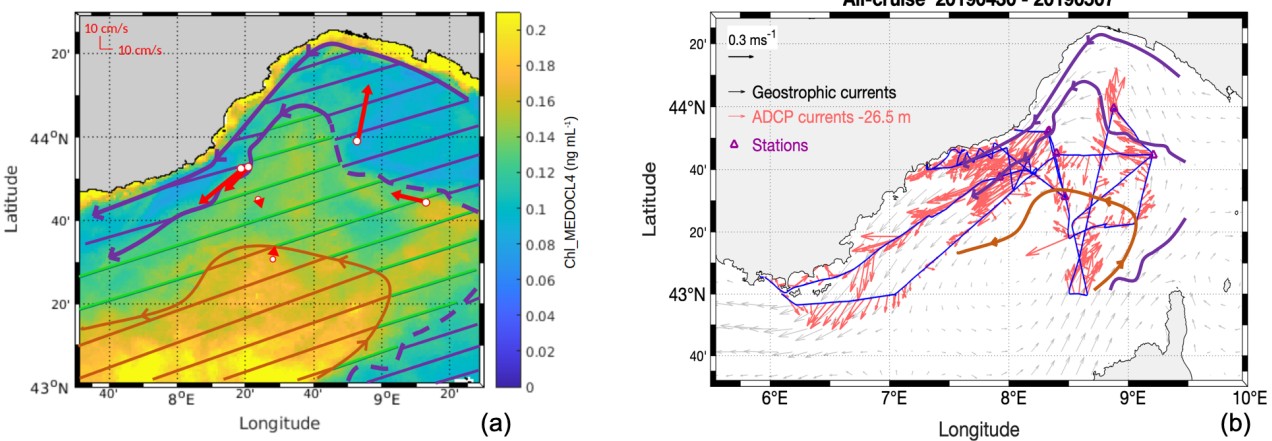

**Figure 3.** (a) Satellite chla averaged concentration (Chl_MEDOCL4, $\text{ng mL}^{-1}$) from 1 to 6 May 2019, used to set the drawn hatched boundaries between the hydrodynamic zones, superimposed with horizontal velocities (VM-ADCP, red vectors) at the stations, averaged over $25 - 150$ m. (b) ADCP horizontal currents at $26.5$ m-depth superimposed on surface geostrophic currents from satellite altimetry.

## 3.2 Storm

During the cruise, an episode of particularly intense winds hit the south of France and the Ligurian Sea. In particular, the Ligurian Sea was exposed to two main winds: NW (Mistral wind) with intensities ranging between 25.8 and $36.1 \, \text{m s}^{-1}$, and N (Tramontana wind) with intensities ranging between 20.6 and $25.8 \, \text{m s}^{-1}$. In this zone, this episode began during the night between 4 and 5 May 2019, reached its maximum intensity on 5 May around 5 am, and finished on 5 May in the evening.

Although the conjunction of these two winds is a classical situation in the Ligurian Sea, this event was particularly intense.

After sheltering during the storm, the ship came back to the storm zone during the night between 5 and 6 May. The model shows that during the storm maximum (5 May around 5 am), the ship-sampled zone (marked with squares in Fig. 4) was affected by a wind intensity peak of $26\,\mathrm{m\,s^{-1}}$ associated with an intense negative net heat flux of $-400\,\mathrm{W\,m^{-2}}$. This sampled zone was in the core of a corridor area ($8°$ E 42.5-44.5° N) with strong wind intensities and high negative heat fluxes (Fig. 4).

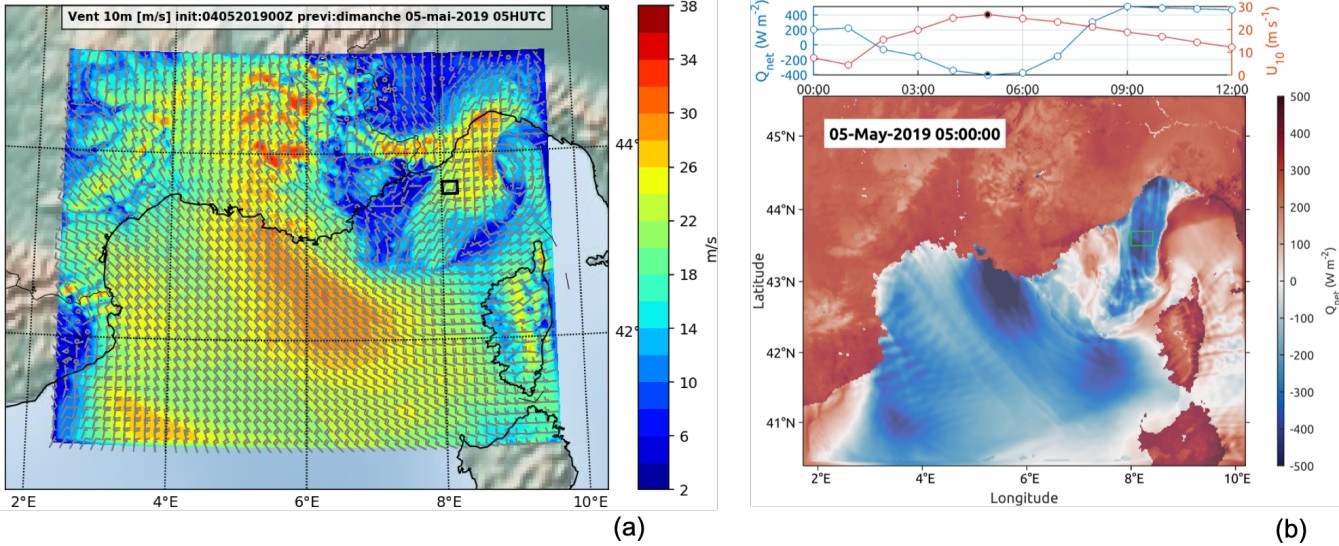

(a)                                                                                    (b)

**Figure 4.** Results of the wind situation on 5 May (WRF model WRF-ARW v4.2.1). The squared areas (black and green) identify the TSG region of interest sampled one day after the storm. (a) Wind intensity at $10\,\mathrm{m}$ on 5 May, 05:00. (b) Heat flux on 5 May, 5:00 (bottom). Temporal distribution of wind intensity and heat flux on 5 May between 0:00 and 12:00 (top) in the green squared area.

### 3.3 Surface hydrodynamics and hydrology

The general properties of the surface waters include surface conservative temperature, absolute salinity, chla concentration (Chl_tsg and Chl_insitu), and in situ Nitrate ($NO_3^-$) concentration (Fig. 5, 6). The conservative temperature was globally warmer near the coast and in the NC (mean value of 15.7°C in the NC), and cooler in the intermediate and recirculation zone (mean value of 15.4°C in the recirculation zone). The absolute salinity was lower near the coast and in the NC (mean value of

230 $38.12\,\mathrm{g\,kg^{-1}}$ in the NC), and higher in the intermediate and recirculation zone (mean value of $38.38\,\mathrm{g\,kg^{-1}}$ in the recirculation zone). The TSG chla (Chl_tsg) concentration mean value was $0.29\,\mathrm{ng\,mL^{-1}}$ over the whole cruise, with a lower mean value in the NC ($0.21\,\mathrm{ng\,mL^{-1}}$) than in the recirculation zone ($0.33\,\mathrm{ng\,mL^{-1}}$).

When the ship came back offshore less than 24 h after the maximum storm intensity, we observed a patch of low-temperature ($< 14.8°C$) and high-salinity ($> 38.28\,\mathrm{g\,kg^{-1}}$) water, with a sharp horizontal gradient separating it from surrounding waters (Fig. 5). This patch was associated with an increase in mean chla: Chl_insitu rose to $0.65\,\mathrm{ng\,mL^{-1}}$ while the mean value for the whole cruise was $0.25\,\mathrm{ng\,mL^{-1}}$. Similarly, Chl_tsg maximal value inside the patch was of $1.11\,\mathrm{ng\,mL^{-1}}$. The nutrients also showed an increase, in particular the $NO_3^-$ concentration which was up to $1.25\,\mu\mathrm{M}$, for a mean value of $0.15\,\mu\mathrm{M}$ for the whole cruise (Fig. 6b). This particular zone of interest is highlighted in cyan in Fig. 5 and 6 and corresponds to longitudes between $8°\,\mathrm{E}$ and $8°15'\,\mathrm{E}$ and latitudes between $43°33'\,\mathrm{N}$ and $43°42'\,\mathrm{N}$.

A TS diagram was used to describe the water masses (Fig. 7a). The water masses were classified using the absolute salinity S and the conservative temperature $\Theta$, from black for deeper and denser waters ($S \geq 38.61\,\mathrm{g\,kg^{-1}}$) to lighter orange/yellow tones for the shallower ones ($S < 38.61\,\mathrm{g\,kg^{-1}}$). Hence surface waters included mostly yellow waters ($S \leq 38.46\,\mathrm{g\,kg^{-1}}$ for $\Theta \leq 13.8°C$, and $S \leq 38.38\,\mathrm{g\,kg^{-1}}$ for $\Theta > 13.8°C$) and orange waters ($38.38\,\mathrm{g\,kg^{-1}} < S \leq 38.62\,\mathrm{g\,kg^{-1}}$ & $\Theta > 13.8°C$). As can be seen in Fig. 7b, the yellow waters were present at the surface in the NC area and the intermediate zone and will be thereafter named "NC waters". Conversely, the orange waters were localised at the surface offshore in the recirculation zone of the basin-scale cyclonic circulation and will be thereafter called "recirculation waters".

A cold surface water patch was encountered after the storm in the geographical cyan area in Fig. 5, and in addition by the glider, during the storm in its ascending route (Fig. 11a). The characteristics of this cold surface water patch ($38.31\,\mathrm{g\,kg^{-1}} \leq S \leq 38.45\,\mathrm{g\,kg^{-1}}$ for $14°C \leq \Theta \leq 14.5°C$ and $38.28\,\mathrm{g\,kg^{-1}} \leq S \leq 38.38\,\mathrm{g\,kg^{-1}}$ for $14.5°C \leq \Theta \leq 14.78°C$) are superimposed in cyan on the TS diagram. They correspond to either NC or recirculation waters, with a density anomaly around $28.37 - 28.70\,\mathrm{kg\,m^{-3}}$, and are present around $30 - 40\,\mathrm{m}$ depth before the storm, as can be seen in Fig. 8a. Between $43°\,31'\,\mathrm{N}$ and $43°\,39'\,\mathrm{N}$, these waters have been detected between $50\,\mathrm{m}$ and the surface, by both the MVP during its $7^\mathrm{th}$ transect (after the storm) and the glider at the end of its ascending route during the storm (Fig. 8b). These waters, thereafter called "newly-mixed waters", were present up to the surface in a very localised spot in space and time (Fig. 7c), and are represented in cyan through the paper. The vessel crossed these surface newly-mixed waters on 6 May between 2:32 am and 2:53 am, 3:03 am and 4:03 am, and 5:32 am and 11:36 am, with the vessel moving in and out of these waters. The glider encountered the surface newly-mixed waters on its way North around 10 am on 5 May. It was at this time at about $85\,\mathrm{km}$ from the ship and stayed in these waters until its recovery on the morning of 6 May.

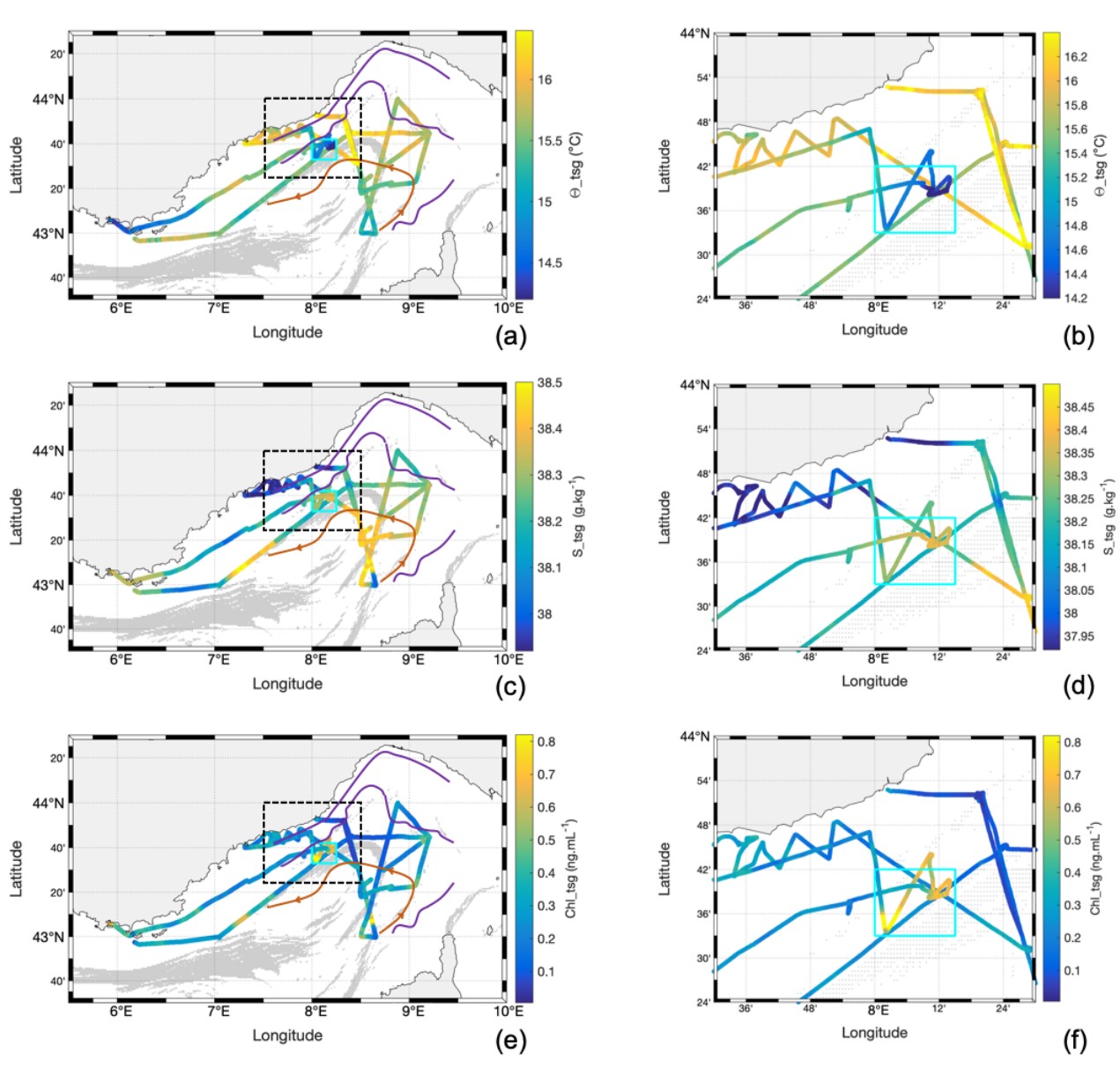

**Figure 5.** Water surface characteristics from TSG along the cruise, superimposed with FSLE calculated from altimetry. (a) (b) Θ_tsg. (c) (d) S_tsg. (e) (f) Chl_tsg concentration. The left panels show the whole geographic region of the cruise and the right panels illustrate the zoom of the indicated region (black dotted square), identifying the particular TSG region of interest (cyan square) sampled one day after the storm.

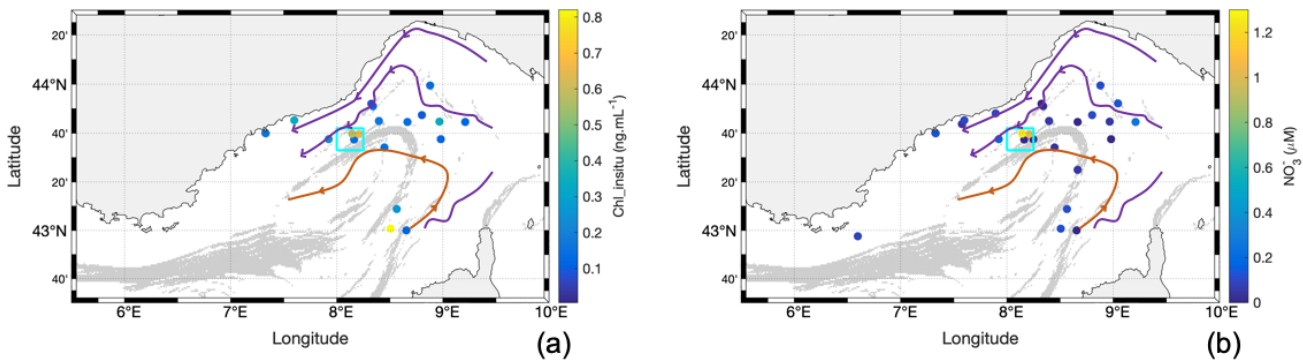

**Figure 6.** Water surface characteristics from discrete in situ sampling along the cruise. (a) Chl_insitu concentration. (b) $NO_3^-$ concentration. The cyan square identifies the particular TSG region of interest sampled one day after the storm.

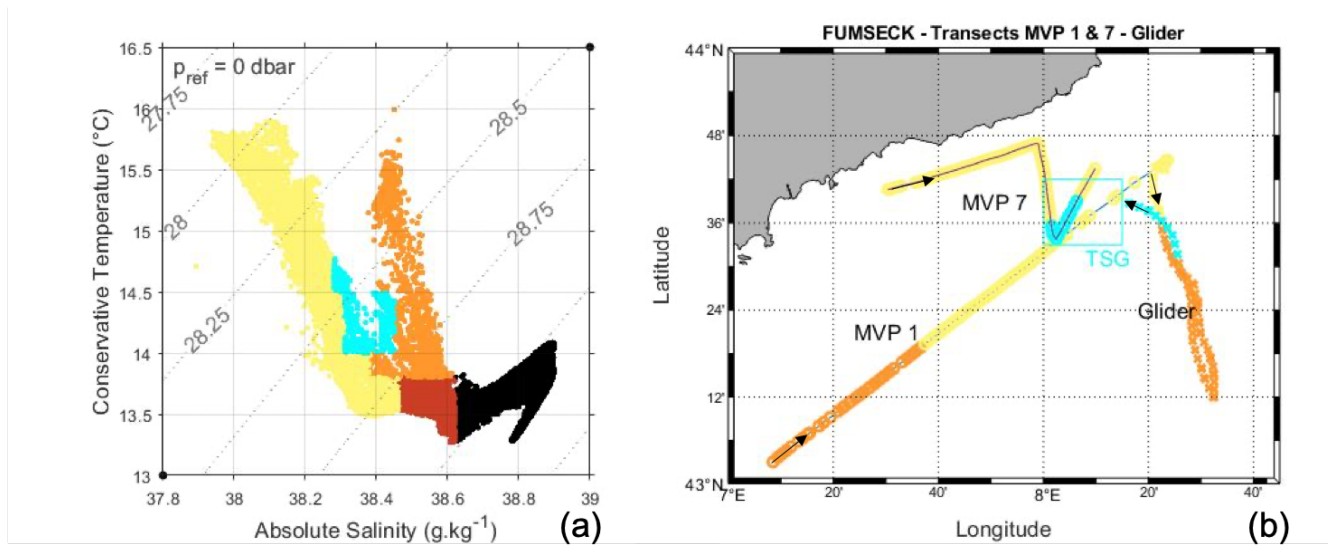

**Figure 7.** Water masses types measured by the MVP (MVP 1 from 30 April 21:29 to 1 May 7:50, and MVP 7 from 5 May 19:22 to 6 May 5:06) and the glider (descending southward from 1 May 8:50 to 04 May 0:29, and ascending northward from 4 May 0:29 to 6 May 3:42). (a) TS diagram from MVP 7 and ascending glider data. (b) Map with the surface colored waters measured by MVP and glider, and TSG zone of interest. The colors are as follows: recirculation waters in orange, NC waters in yellow, and newly-mixed waters in cyan.

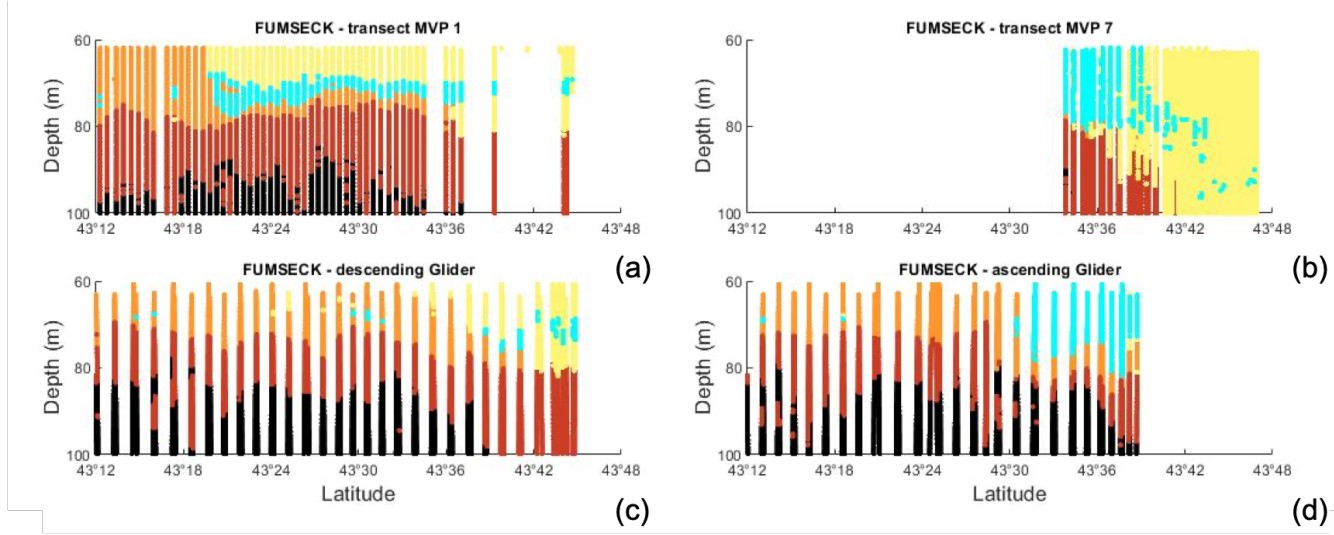

**Figure 8.** Vertical transects versus longitude with associated colored waters. (a) MVP 1, (b) MVP 7, (c) descending glider (southward transect), (d) ascending glider (northward transect).

### 3.4 Chlorophyll-a and total biomass

Chl_insitu varied between 0.07 and 0.82 $\mathrm{ng\,mL^{-1}}$ with a mean $\pm$ sd of $0.25 \pm 0.21\,\mathrm{ng\,mL^{-1}}$, with 20 samples collected all along the cruise (Fig. 6a, Fig. 9a). The standard deviations are representative of the spatiotemporal variability, not the measure-
ment errors. Chl_cyto values followed a similar trend with minimal and maximal values of 0.03 and 0.94 respectively, and a mean $\pm$ sd of $0.26 \pm 0.16\,\mathrm{ng\,mL^{-1}}$ (Fig. 9a). Chl_tsg varied between undetectable values and $1.11\,\mathrm{ng\,mL^{-1}}$, with a mean of $0.29 \pm 0.16\,\mathrm{ng\,mL^{-1}}$ and a mean spatial resolution of $0.16\,\mathrm{km}$ with a total of 8385 points (Fig. 5e, Fig. 9b).

Ocean color chla match-ups with Chl_cyto (Fig. 9b) were significantly higher for Chl_ACRI than for Chl_MEDOCL4
($0.27 \pm 0.07$ and $0.15 \pm 0.05\,\mathrm{ng\,mL^{-1}}$, p < 0.001, block-bootstrap test (Appendix D)). Maximal values of Chl_ACRI and Chl_MEDOCL4 (0.48 and 0.51 $\mathrm{ng\,mL^{-1}}$, respectively) were below the maximal values of Chl_cyto and Chl_tsg.

Total biomass of phytoplankton ranged between 13.75 and 77.94 $\mathrm{ngC\,mL^{-1}}$ with a mean $\pm$ sd of $33.05 \pm 11.23\,\mathrm{ngC\,mL^{-1}}$ and followed Chl_cyto trends with a correlation of 0.52 (n $=$ 400) when considering the entire data set, and of 0.72 (n $=$ 382)
when removing the data from the newly-mixed waters. For the newly-mixed waters, the correlation was 0.78 (n $=$ 21).

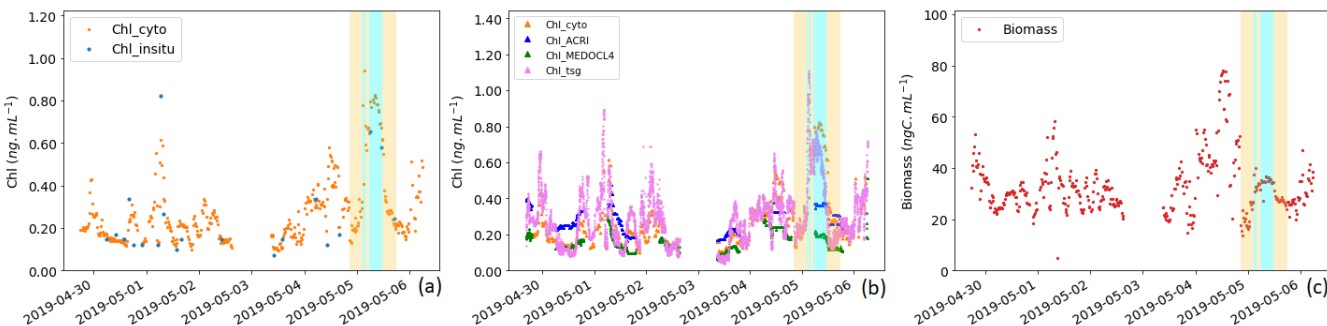

**Figure 9.** (a) Comparison between the surface chla concentration measured in situ (Chl_insitu, $\mathrm{ng\,mL^{-1}}$) and the chla concentration estimation obtained from AFCM (Chl_cyto, $\mathrm{ng\,mL^{-1}}$). (b) Comparison between the surface chla concentration estimated by AFCM (Chl_cyto), ACRI (Chl_ACRI), MEDOCL4 (Chl_MEDOCL4), and fluorometer (Chl_tsg). (c) Total surface phytoplankton biomass variation through the cruise ($\mathrm{ngC\,mL^{-1}}$). The periods corresponding to the surface crossing of the newly-mixed waters (in cyan) and the surrounding NC ones (6 hours before and after the newly-mixed ones, in yellow) are indicated.

## 3.5 Phytoplankton groups and reaction

The most abundant group was the Orgpicopro, followed by the Rednano, Redpicoeuk, Orgnano, and Redmicro (see Table 2). Inversely, the Rednano biomass was the highest, followed by the Orgpicopro biomass. Redpicoeuk biomass was the lowest. Chlorophyll per group per unit of volume regarding the overall study area was also the highest for the Rednano followed by the Orgpicopro. The biomass/Chl_cyto ratio was above 127 for all phytoplankton groups when considering the entire study area.

For all phytoplankton groups except for Orgpicopro, abundances and biomass per unit of volume were twice higher in newly-mixed waters (cyan in Fig. 7a) compared to NC surrounding waters (yellow in Fig. 7a) as shown in Table 2 and Fig. 10. All groups had higher chla values in the newly-mixed waters (Table 2). Conversely, Rednano and Redpicoeuk estimated average sizes were higher with a concomitant higher biomass per cell in the NC surrounding waters than in the newly-mixed ones (Table 2). The biomass/Chl_cyto ratios were lower in NC waters and even more in newly-mixed waters compared to the overall area (see Fig. 13) for all groups, despite lower carbon content per cell. In short, the newly-mixed waters evidenced higher abundances and higher chla concentration and biomass per unit of volume but smaller sizes and biomass per cell (mainly for the Redpicoeuk and Rednano).

**Table 2.** Mean ± standard deviation values for surface abundance, size (equivalent spherical diameter ESD), biovolume per cell, biomass per cell, chla per unit of volume (Chl_cyto), biomass per unit of volume and the ratio biomass over Chl_cyto for the overall sampling waters (n = 400), the NC surrounding waters (n = 20) and the newly-mixed waters (n = 43) (see Fig. 7) for the five AFCM phytoplankton groups identified. The surrounding NC waters correspond to the surface NC waters acquired 6 hours before and after the ship entered newly-mixed waters. A Moving Blocks Bootstrap test between NC surrounding and newly-mixed waters reveals significant differences, bold values are significantly different at a Bonferroni-corrected 5% level.

| Observable | Waters | Orgpicopro mean ± SD | Redpicoeuk mean ± SD | Rednano mean ± SD | Orgnano mean ± SD | Redmicro mean ± SD |
|---|---|---|---|---|---|---|
| Abundance (cell mL$^{-1}$) | Overall | 51556 ± 21827 | 1017 ± 473 | 3686 ± 887 | 211 ± 192 | 3 ± 2 |
| | NC surrounding | 63239 ± 29087 | **1175 ± 397** | **2746 ± 546** | **160 ± 84** | **4 ± 2** |
| | Newly-mixed | 61162 ±4898 | **2334 ± 392** | **4597 ± 333** | 325 ± 34 | **6 ± 1** |
| Size (ESD, $\mu$m) | Overall | 0.98 ± 1.02 | 2.18 ± 1.76 | 3.30 ± 2.45 | 5.22 ± 4.50 | 11.38 ± 10.72 |
| | NC surrounding | 0.96 ± 0.97 | **2.14 ± 1.69** | **3.18 ± 2.35** | 4.91 ± 4.40 | 9.98 ± 8.46 |
| | Newly-mixed | 0.94 ± 1.02 | **1.92 ± 1.61** | **3.02 ±2.29** | 4.83 ± 4.45 | 9.64 ± 8.16 |
| Biovolume ($\mu$m$^3$) | Overall | 0.50 ± 0.56 | 5.53 ± 2.95 | 19.16 ± 7.91 | 75.24 ± 47.90 | 1051.26 ± 1586.96 |
| | NC surrounding | 0.47 ± 0.50 | **5.24 ± 2.58** | **16.97 ± 6.86** | 62.6 ± 45.00 | 585.94 ± 566.35 |
| | Newly-mixed | 0.45 ± 0.57 | **3.72 ± 2.18** | **14.54 ± 6.32** | 59.33 ± 46.46 | 470.97 ± 286.68 |
| Biomass/cell (pgC cell$^{-1}$) | Overall | 0.14 ± 0.16 | 1.13 ± 0.66 | 5.52 ± 2.57 | 18.00 ± 12.20 | 165.31 ± 216.90 |
| | NC surrounding | 0.14 ± 0.14 | **1.08 ± 0.59** | **4.98 ± 2.28** | 15.36 ± 11.56 | 103.26 ± 90.60 |
| | Newly-mixed | 0.13 ± 0.16 | **0.80 ± 0.81** | **4.36 ± 0.12** | 14.68 ± 11.82 | 87.70 ± 58.49 |
| Chl_cyto (ng mL$^{-1}$) | Overall | 0.061 ± 0.051 | 0.006 ± 0.006 | 0.184 ± 0.104 | 0.014 ± 0.015 | 0.003 ± 0.002 |
| | NC surrounding | **0.100 ± 0.600** | **0.009 ± 0.005** | **0.173 ± 0.083** | **0.012 ± 0.001** | **0.004 ± 0.002** |
| | Newly-mixed | **0.160 ± 0.014** | **0.025 ± 0.005** | **0.526 ± 0.079** | **0.032 ± 0.000** | **0.006 ± 0.001** |
| Biomass (ngC mL$^{-1}$) | Overall | 7.47 ± 3.97 | 1.12 ± 0.43 | 20.30 ± 5.58 | 3.77 ± 3.75 | 4.81 ± 4.82 |
| | NC surrounding | 7.72 ± 2.71 | **1.22 ± 0.29** | **13.58 ± 2.18** | **2.38 ± 1.09** | **3.77 ± 1.94** |
| | Newly-mixed | 7.90 ± 3.61 | **1.86 ± 0.24** | **20.05 ± 1.39** | **4.77 ± 4.81** | **5.34 ± 1.26** |
| Biomass/Chl_cyto | Overall | 158.10 ± 56.3 | 268.4 ± 99.2 | 127.4 ± 43.2 | 292.1 ± 71.6 | 206.4 ± 202.5 |
| | NC surrounding | **86.0 ± 26.3** | **158.0 ± 40.9** | **87.5 ± 21.5** | **218.5 ± 33.2** | 115.6 ± 94.2 |
| | Newly-mixed | **48.6 ± 3.8** | **76.8 ± 9.5** | **38.6 ± 4.3** | **148.4 ± 9.9** | 93.6 ± 142.0 |

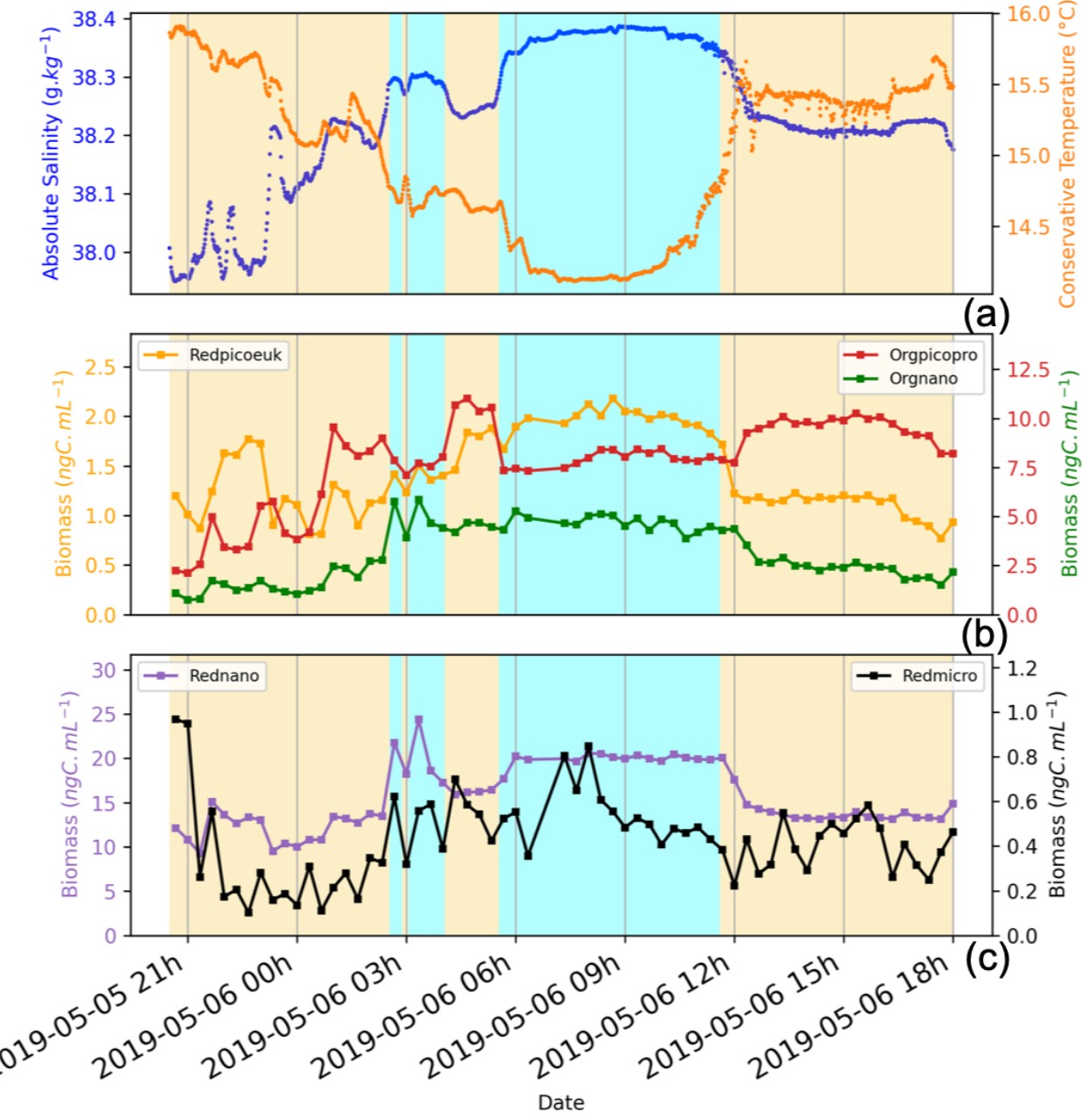

**Figure 10.** Illustration of the newly-mixed waters (corresponding to the cyan background) and their direct surroundings (NC waters, corresponding to the yellow background), in terms of surface temperature, salinity, and biomass per phytoplankton group $(\mathrm{ngC\,mL^{-1}})$. (a) Variation of the surface Absolute Salinity (blue dots) and surface Conservative temperature (orange dots). (b) Variation of the surface biomass for Redpicoeuk (Orange line), Orgpicopro (Red line), and Orgnano (Green line). (c) Variation of the surface biomass for Rednano (violet line) and the Redmicro (black line). The vertical axis tick colors indicate the associated curve. Similarly, ticks's labels and titles written in two different colors indicate that two curves are associated with the same axis.

### 3.6 Subsurface fluorescence signal observed by the glider

Referring to the surface water masses of section 3.3, the glider entered the newly-mixed surface waters on its northward return transect (5 May), leaving behind recirculation waters (Fig. 7c). Down to approximately $60\,\mathrm{m}$ depth, the surface temperature and salinity steeply decreased (see Fig. 11), moving from recirculation waters to newly-mixed waters. The fluorescence near the surface increased rapidly by a factor of four (Fig. 12b) as the mixed layer depth recorded by the glider deepened from 15 to $50\,\mathrm{m}$ (Fig. 12a). However, the integrated fluorescence content in the upper $100\,\mathrm{m}$ did not show any noticeable variation (Fig. 12b).

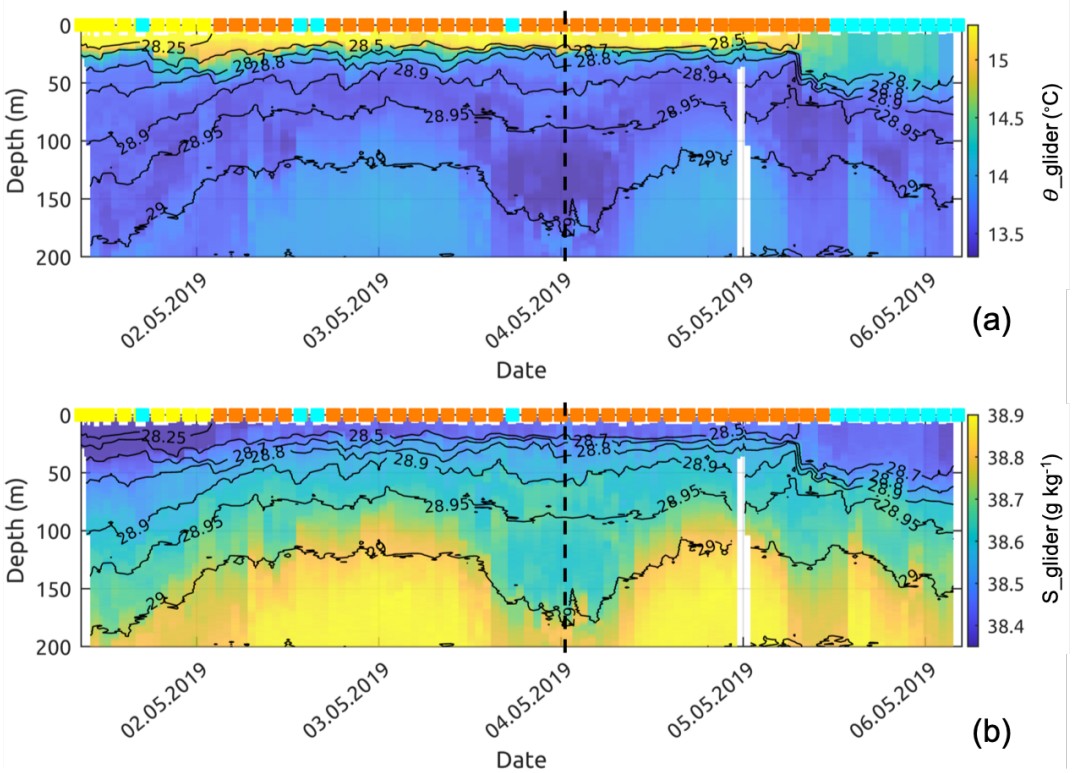

**Figure 11.** Glider profiles of (a) conservative temperature and (b) absolute salinity. The colored squares correspond to the dominant water mass according to Fig. 7 observed at $10\,\mathrm{m}$ depth by the glider. The dashed vertical line represents the time separating the descending and ascending transects.

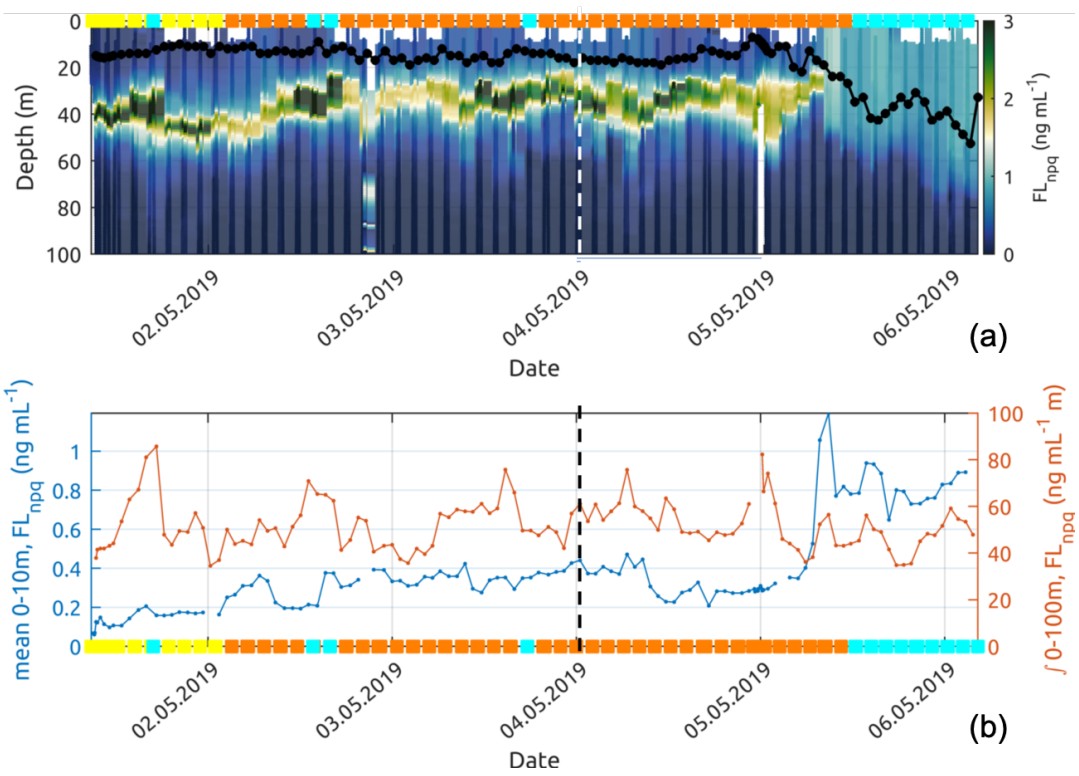

**Figure 12.** (a) Fluorescence observed by the glider and corrected from non—photochemical quenching following Xing et al. (2012). The black line with dots shows the mixed layer depth (MLD) at each glider profile. (b) Near-surface ($0 - 10\,\mathrm{m}$ average) and integrated over $0 - 100\,\mathrm{m}$ chla fluorescence concentration along the glider track. The colored squares correspond to the dominant water mass according to Fig. 7 observed at $10\,\mathrm{m}$ depth by the glider. On both plots, the dashed vertical lines represent the time separating the descending and ascending transects.

## 4 Discussion

In the NW Mediterranean Sea, in May, the water column is generally well stratified with nearly undetectable surface nutrient
availability (Pasqueron De Fommervault et al., 2015). This was indeed the oceanographic setting before an intense storm dominated by north-westerly winds impacted the water column. The analysis of 30 years of coastal data in the South of France (Toulon and Marignane) by Meteo France shows that the typical periods of intense wind occur at the end of winter and middle of autumn. In Toulon, winds of intensity $> 27.8\,\mathrm{m\,s^{-1}}$ occur on average 8 times per year, but only once every 4 years in May; winds of intensity $> 36.1\,\mathrm{m\,s^{-1}}$ occur on average once every 2 years, and once every 30 years in May. The total occurrences of different wind intensities for the whole 1981-2010 period are shown in Table 3. The wind intensity of the studied storm, reaching a maximum of $36.1\,\mathrm{m\,s^{-1}}$, was rare in the Mediterranean Sea, and similar to the average wind intensity of the typhoons studied by Wang (2020).

**Table 3.** Occurences of different wind intensities events for the 1981-2010 period in the South of France, in total and in May only (http://tempetes.meteo.fr/spip.php?article221).

| Location \ Wind intensity | | $> 27.8\,\mathrm{m\,s^{-1}}$ $(> 100\,\mathrm{km\,h^{-1}})$ | $> 30.6\,\mathrm{m\,s^{-1}}$ $(> 110\,\mathrm{km\,h^{-1}})$ | $> 33.3\,\mathrm{m\,s^{-1}}$ $(> 120\,\mathrm{km\,h^{-1}})$ | $> 36.1\,\mathrm{m\,s^{-1}}$ $(> 130\,\mathrm{km\,h^{-1}})$ |
|---|---|---|---|---|---|
| Toulon | Tot | 232 | 96 | 36 | 14 |
| | May | 8 | 1 | 1 | 1 |
| Marignane | Tot | 205 | 60 | 13 | 5 |
| | May | 6 | 0 | 0 | 0 |

The physical and biogeochemical data, collected thanks to the deployment of high-resolution sensors, showed a clear shift in the local ocean physical-biological conditions after the storm. These changes included a steep change in temperature and salinity, and increases in surface chla concentrations and surface phytoplankton biomass and abundances.

Overall, the abundances of Redpicoeuk and Rednano were more than twice higher in April during the MERITE-HIPOCAMPE cruise (Boudriga et al., 2022) than during the FUMSECK cruise, suggesting that the FUMSECK cruise occurred after the spring bloom events when nutrients in the euphotic layer are consumed. Only Orgpicopro, related to *Synechococcus* cells, were similar for both samplings. Abundances of Orgnano, Redpicoeuk, and Rednano were close to the ones in the Eastern Mediterranean Sea in May (Latasa et al., 2022). Conversely, the abundances of phytoplankton groups during FUMSECK were on average twice higher than the ones observed at the same location during the OSCAHR cruise in November 2015 (Marrec et al., 2018) for the Rednano and the Orgpicopro but similar for the Redpicoeuk. The size of Rednano was smaller on average (-20%) than the one observed during the OSCAHR cruise but larger for the Redpicoeuk (+30%) and the Orgpicopro (+20%).

The conversion into chla from the total red fluorescence evidenced Rednano as the main contributor during the entire study. The same observation held in terms of biomass. All groups exhibited higher Chla_cyto in the newly-mixed waters with respect to the surrounding waters, reaching up to +68% for Rednano in the newly-mixed waters compared to the NC surroundings, despite the cells being smaller. Similarly, fluorescence per group was much higher in the cold core of the OSCAHR eddy than in the surrounding warm water. During OSCAHR, the ratio in chla reached 1.5 between the cold and the warm waters while in our study, Chl_cyto for Rednano and for Redpicoeuk was nearly 3 times higher in the cold newly-mixed waters. This suggests that the cells did not have time to photoacclimate, or that different species were involved. Indeed, the newly-mixed water was sampled less than one day after deeper layers reached surface layers.

The phytoplankton abundances and size classes distribution inform on the capacity of the area to sustain the marine food web, while the carbon/chla ratio is an indicator of photoacclimation status, especially interesting when primary production is calculated from chla only (Behrenfeld et al., 2002). It also gives insight into the rapid changes in light conditions, as some time is needed to photoacclimate, and adjust the pigment content of a cell to the new light conditions (Lewis et al., 1984). Most of the carbon over chlorophyll estimates are bulk, only few studies attempted to convert values per size class. This paper aims at contributing to the estimated ratio from field studies with much higher precision thanks to the clear separation between phytoplankton and bulk particulate organic carbon given by AFCM, and also because of the resolution at the single cell level. The estimation of the cell carbon biomasses could be biased by the errors made during the prior estimation of the cell biovolumes, but also by the use of biovolume-to-biomass conversion factors from the literature. Nevertheless, the high variability of the carbon/chla values between phytoplankton groups evidenced different metabolisms between groups, Redpicoeuk having a much higher ratio (268.4) than Rednano (127.4). The Redpicoeuk carbon/chla ratio range outside the newly-mixed waters was similar to the highest values found in summer in the study of Calvo-Díaz et al. (2008), where values for picoplankton varied seasonally from 0.07 to 282. Inside the newly-mixed waters, Redpicoeuk carbon/chla values were similar to the ones in the North European seas in summer, under higher nutrient environments with lower light conditions (maximal values were close to 85 in open waters, (Jakobsen and Markager, 2016)). The carbon/chla ratio integrating all groups varied from approx. 90 to 250 in surface conditions but dropped down to 50 in the newly-mixed waters (Fig. 13). The high ratios observed before the storm could reflect the high light and low nutrient conditions of the post-bloom oligotrophic period sampled in the Ligurian Sea. The remarkable drop in the ratio observed in the cold water patch could be a signature of a sudden change in phytoplankton cells, that may have translated the not-yet photoacclimated configuration of the cells to high light conditions (Jakobsen and Markager, 2016).

While surface observations alone suggested a rise in chla concentrations (Fig. 5 and 6), the integrated chla values from the glider fluorometer rather suggested that this surface increase is due to a dilution of the deep chlorophyll maximum in the mixed layer during the storm (Fig. 12). The deepening of the mixed layer depth can lead to the dilution - by vertical mixing - of phytoplankton cells previously concentrated in the deep chlorophyll maximum.

Typhoons can be compared to the type of storm observed in our study by the intensity and duration of the winds triggering a fast decrease of surface temperature and an increase in surface chla. Most of the typhoons enhance chlorophyll surface concentration (Wang, 2020). In open water tropical and sub-tropical areas, the dilution phenomenon of the deep chlorophyll maximum after typhoons was warned to be a source of overestimation of potential phytoplankton production when using only satellite observation, because the nitracline is not always affected (Chai et al., 2021), and because chla from the deep chlorophyll maximum is not always related to higher biomass and production (Marañón et al., 2021). The increase in chla after the deepening of the mixed layer depth in post-bloom periods entailed by wind events is not obvious as demonstrated by Andersen and Prieur (2000). In our case, the deepening of the mixing due to the storm was accompanied by an increase in surface nutrients that could only be linked to the uplift of the nitracline, as we were far enough from coastal run-off influences. This mixing was related to the spreading and the increasing of the phytoplankton in the upper layer in terms of biomass and chla. In addition, the mixing could lead to a possible dilution of the grazers favouring the pico-nanophytoplankton accumulation in the shallowing mixed layers a few days after (Morison et al., 2019). This could in turn foster the integrated primary production by enhancing phytoplankton division rate and biomass (Behrenfeld, 2010) which, when grazers are diluted, is related to higher organic carbon export efficiency (Henson et al., 2019). This phenomenon was also observed after winter storms in the Sargasso Sea, where diatoms increase was maximal within two days after shoaling of the mixed layer depth (Krause et al., 2009). These pulsed production events could be responsible for up to 20% of the global primary production in the Sargasso Sea (Lomas et al., 2009).

Our observations captured the short-term physical and phytoplankton response to a storm, with rapid and strong changes observed but without the possibility to follow in situ post-conditions. Although not representative of what happens in the entire mixed water column, satellite data showed an effect on surface temperature and chla within the ship-glider storm geographical zone (longitudes between $8°$ E and $8°30'$ E and latitudes between $43°30'$ N and $43°42'$ N). In this zone, the mean value of SST was lower during four days following the storm (6-10 May, $14.8°C$) than between the 20 April-20 May period ($15.1°C$), while the mean value of Chl_ACRI was higher ($0.44\,\mathrm{ng\,mL^{-1}}$ with respect to $0.32\,\mathrm{ng\,mL^{-1}}$), suggesting that the pico-nanophytoplankton size classes could have had the time to grow and accumulate, as their growth rate is close to one to two divisions a day when nutrients and light are available (Morison et al., 2019). This is supported by the increase in Particulate Organic Carbon concentration (POC, https://oceancolor.gsfc.nasa.gov/atbd/poc/#sec_6, MODIS Aqua L3 product) in the considered zone. Indeed, a higher value of POC concentration is observed during the four days following the storm (6-10 May, $104.1\,\mathrm{ng\,mL^{-1}}$) compared to the 20 April-20 May period ($88.7\,\mathrm{ng\,mL^{-1}}$), suggesting that the whole trophic chain may be impacted by the storm.

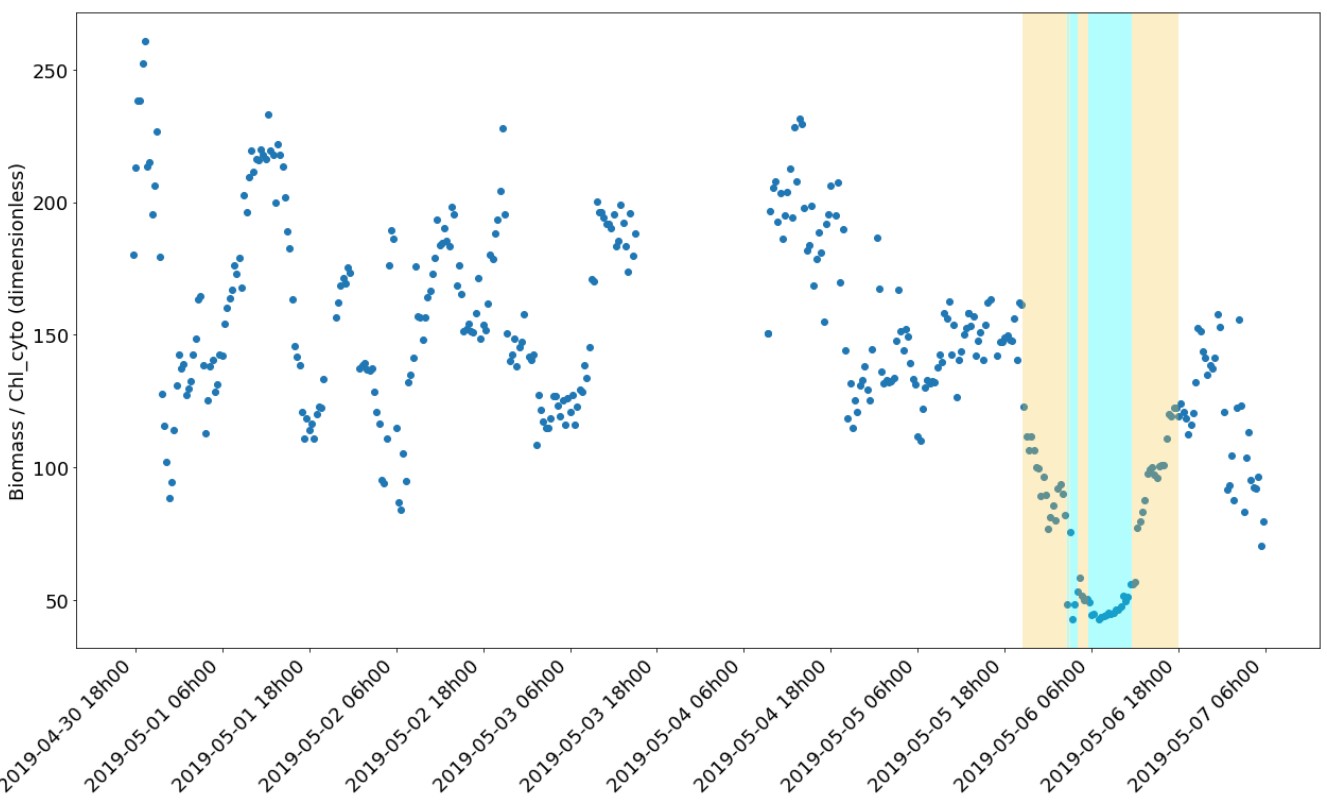

**Figure 13.** Evolution of the Biomass $(\mathrm{ngC\,mL^{-1}})$ / Chl_cyto $(\mathrm{ng\,mL^{-1}})$ ratio through the cruise. The yellow and cyan color spans correspond to the water masses of Fig. 10.

## 5 Conclusion

During the FUMSECK cruise, the deployment of high spatio-temporal resolution instruments has made it possible to observe the link existing between the fine-scale physical structures and phytoplankton size-class distribution in the Ligurian Sea. Initially, the studied area was under typical post-bloom physical and biological characteristics of the NW Mediterranean Sea with surface stratified conditions, a MLD at about $15\,\mathrm{m}$ and close to undetectable surface concentrations of chlorophyll, where cells $< 4-5\,\mu\mathrm{m}$ dominated biomass. A storm of high intensity occurred during the cruise period and its effects on the water column and the surface phytoplankton were specifically studied thanks to high-resolution measurements performed simultaneously with a glider, a MVP, a surface thermo-salinometer, a VM-ADCP, and an automated flow cytometer. The in situ dataset was strengthened by satellite and numerical modelling data.

The area affected by the storm was characterised by waters mixed from depths down to $60\,\mathrm{m}$ up to the surface, with a clear dilution of the deep chlorophyll maximum, leading to abrupt changes in the phytoplankton abundances in surface waters. The study of phytoplankton at the single-cell level showed clear physiological changes with a drop in carbon/chla ratio associated with an increase in abundances and biomass. These physiological shifts can be regarded as a reaction to the sudden changes. The storm, although identified as a rare event in this area, should be considered an important feature to study within the fine-scale physical-biological coupling, especially in stratified surface oligotrophic conditions where nutrient increases can trigger pulsed production and affect global biogeochemical budgets.

These results pave the way for future oceanic cruises, and in particular for the BioSWOT-Med cruise in 2023. This cruise is planned as part of the "Adopt a Cross Over" initiative organising simultaneous oceanographic cruises around the world during the fast sampling phase of the new satellite SWOT (Surface Water and Ocean Topography) (d'Ovidio et al., 2019), which will allow the precise observation of fine-scale ocean dynamics. The aim is to study the fine-scale features and their influence on biology, with methodology supporting offshore, multi-instrumental, multi-technique, multi-scale, and multi-disciplinary observations.

These results highlight the need for concomitant observations of physics and biology with high spatio-temporal resolution in order to understand the effect of physical forcing events, such as storms, on marine ecosystems. Considering that changes in both the frequency and the intensity of Mediterranean storms are expected (Lionello et al., 2006; Flaounas et al., 2021), these results may help to estimate the impact of climate change on the ecology and biogeochemistry of the Mediterranean Sea.

*Data availability.*

Data are available here:

https://dataset.osupytheas.fr/geonetwork/srv/eng/catalog.search#/metadata/5bda8ab8-79e7-4dec-9bcb-25a3196e2f9a

## 420   Appendix A: Identification of phytoplankton

The phytoplankton groups were identified on dimensionless cytograms by comparing two by two the optical pulse shapes per cell, as collected by the Cytosense flow cytometer (Fig. A1). The groups were manually gated. The names of the groups were decided prior to the publication of Thyssen et al. (2022), and will be coupled to their standard names in the open access dataset.

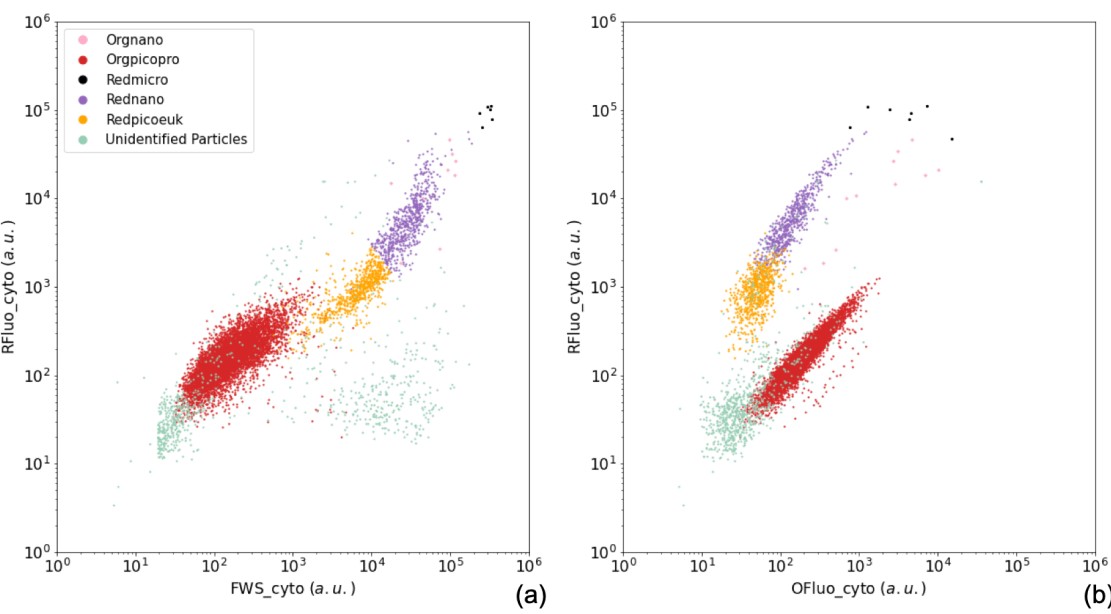

**Figure A1.** Manual identification of the main phytoplankton functional groups. Two-dimension cytograms representing: (a) the area under the curve of red fluorescence (RFluo_cyto, (a.u.)) versus forward scatter (FWS_cyto (a.u.)) of each particle, depicting the main cytometric functional groups identified, namely Orgnano (pink dots), Orgpicopro (red dots), Redmicro (black dots), Rednano (purple dots), Redpicoeuk (orange dots) and the Unidentified particles group (green dots). (b) the area under the curve of red fluorescence (RFluo_cyto (a.u.)) versus orange fluorescence (Ofluo_cyto (a.u.)) of each particle, evidencing the same groups. The size of the Orgnano and Redmicro points located in the most top right part of the cytograms was increased in order for them to be visible as these groups count very few cells in each sample.

## Appendix B: Chla agreement between the glider and the TSG

No pre-cruise calibration of the glider Ecopuck was carried out. Nevertheless, we observe a good statistical agreement between the measurements of the glider with those taken from the ship TSG (Fig. B1). Over a sample of $N = 33$ glider profiles where the glider-ship distance is lower than $40\,\mathrm{km}$, surface chla from the ship TSG (Chl_tsg) and chla $0 - 10\,\mathrm{m}$ average from the glider have a correlation coefficient of $R = 0.76$ (with a significant p-value of $2.5\,10^{-7}$) and a mean standard error of $0.067\,\mathrm{ng\,mL^{-1}}$, well below of the amplitude of the observed signal during the storm (maximum Chl_tsg of $1.11\,\mathrm{ng\,mL^{-1}}$). Values from the

onset of the storm have been excluded (grey values after 5 May) since the glider was experiencing different conditions than the ship sheltering from the bad weather. At the end of the time series, when the glider was recovered, the values between the two platforms agree well again, which gives a good confidence in the chla fluorescence signals described by the glider's sensor during its mission.

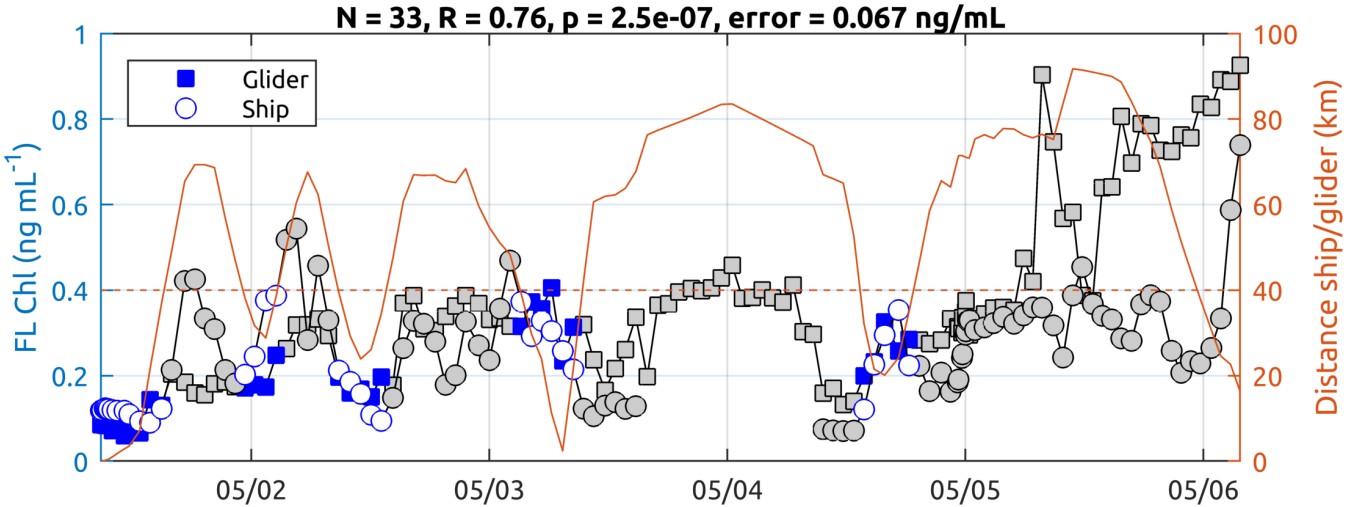

**Figure B1.** Comparison of chla between the ship TSG and the $0 - 10\,\mathrm{m}$ average from the glider. The numbers indicated on top of the figure and the blue markers correspond to measurements where the glider-ship distance is lower than $40\,\mathrm{km}$.

## Appendix C: Details on the satellite products

The satellite products exploited for FUMSECK are the following (Barrillon et al., 2020):

- SSH and associated geostrophic currents
  - "Mediterranean ocean gridded L4 Sea Surface Heights and derived variables" (SEALEVEL_MED_PHY_L4_NRT_OBSERVATIONS_008_050, now SEALEVEL_EUR_PHY_L4_NRT_OBSERVATIONS_008_060, https://resources.marine.copernicus.eu/product-detail/SEALEVEL_EUR_P HY_L4_NRT_OBSERVATIONS_008_060/INFORMATION): $0.125° \times 0.125°$, multi-satellite.

– SST

- "Mediterranean Sea - High resolution and ultra high resolution L3S Sea Surface Temperature" (SST_MED_SST_L3S_NRT_OBSERVATIONS_010
_012, https://resources.marine.copernicus.eu/product-detail/SST_MED_SST_L3S_NRT_OBSERVATIONS_010_012/INFORMATION):
$0.01° \times 0.01°$, strict temporal window (local nighttime) to avoid diurnal cycle and cloud contamination, provides supercollated (merged multisensor,
L3S), SST data remapped over the Mediterranean Sea.

- "Mediterranean Sea high resolution and ultra high resolution Sea Surface Temperature analysis" (SST_MED_SST_L4_NRT_OBSERVATIONS_010
_004, , https://resources.marine.copernicus.eu/product-detail/SST_MED_SST_L4_NRT_OBSERVATIONS_010_004/INFORMATION):
$0.01° \times 0.01°$, nighttime images, multi-satellite.

     – Chla

- "Global ocean Chlorophyll from satellite observations" (OCEANCOLOUR_GLO_CHL_L3_NRT_OBSERVATIONS_009_032, now OCEANCO
LOUR_GLO_BGC_L3_NRT_009_101, https://resources.marine.copernicus.eu/product-detail/OCEANCOLOUR_GLO_BGC_L3_NRT_009_101/I
NFORMATION): $4 \, km \times 4 \, km$, ACRI-ST company, multi-satellite, hereafter called Chl_ACRI.
- "Mediterranean Sea surface Chlorophyll concentration from multi satellite observations" (OCEANCOLOUR_MED_CHL_L3_NRT_OBSERVA
TIONS_009_040, now OCEANCOLOUR_MED_BGC_L3_NRT_009_141, https://resources.marine.copernicus.eu/product-detail/OCEANCOLOU
R_MED_BGC_L3_NRT_009_141/INFORMATION): $1 \, km \times 1 \, km$, multi-satellite, hereafter called Chl_MEDOCL3.
- "Mediterranean Sea daily interpolated surface Chlorophyll concentration from multi satellite observations" (OCEANCOLOUR_MED_CHL_L4_NRT
_OBSERVATIONS_009_041, now OCEANCOLOUR_MED_BGC_L4_NRT_009_142, https://resources.marine.copernicus.eu/product-detail/OC
EANCOLOUR_MED_BGC_L4_NRT_009_142/INFORMATION): $1 \, km \times 1 \, km$, multi-satellite, hereafter called Chl_MEDOCL4.

## Appendix D: Testing the mean differences in the phytoplankton groups in different water types

The significance of the differences in means of each phytoplankton group between water types was tested using two-tailed
tests based on the Moving Blocks Bootstrap principle (Liu et al., 1992). Using a bootstrap-based test avoided assuming that the
observations were mutually independent and have to follow Gaussian distributions (given the small sample size in each water
type). These assumptions were indeed violated in our case. Instead, the stationarity of the samples originating from each water
mass was assumed. Sampling the observations by blocks of adjacent observations preserves the serial auto-correlation existing
in the sample. The size of the blocks is in practice left to the practitioner and values in [1,4] were tested and did not influence
the results. The number of bootstrap samples used to perform the tests was 3000 draws. The level of the tests was 5% with a
Bonferroni correction (Dunn, 1961) to account for multiple hypotheses testing.

*Author contributions.*

Jean-Luc Fuda (JLF) prepared the instruments prior to the cruise, and deployed them onboard, together with Stéphanie Barrillon (SB), Andrea Doglioli (SD), Gérald Grégori (GG), Melilotus Thyssen (MT), and Roxane Tzortzis (RT). Anne Petrenko operated SPASSO and analysed the water masses, Caroline Comby analysed the current data. GG and MT prepared and operated the flow cytometer, MT and Robin Fuchs analysed and interpreted the flow cytometry data. Nagib Bhairy prepared and piloted the glider, Frédéric Cyr performed the first treatment of the glider data and Anthony Bosse (AB) analysed the glider data. Christophe Yohia performed the model data, and analysed the model data together with AB. AD analysed the MVP data. Francesco d'Ovidio and AD initiated the project. SB designed the experiment, lead the research, and prepared the manuscript with contributions from all co-authors. All authors contributed to the manuscript.

*Competing interests.*

The authors declare that they have no conflict of interest.

*Acknowledgements.* We thank the captain and the crew of the RV *Téthys II* for the cruise and their help with the deployment of the instruments. All of this research was supported by CNES (BioSWOT project) and by the French National program LEFE (Les Enveloppes Fluides et l'Environnement), FUMSECK-vv project (PIs S. Barrillon and A. Petrenko). The flow cytometer was funded by the CHROME (PI M. Thyssen, funded by the Excellence Initiative of Aix-Marseille University – A*MIDEX, a French "Investissements d'Avenir" program), and the FEDER fundings (PRECYM flow cytometry platform). The authors thanks Nicole Garcia and Patrick Raimbault from the MIO-PACEM platform for the chlorophyll-a and the nutrients analysis. SPASSO is operated with the support of the SIP (Service Informatique de Pythéas) and in particular C. Yohia, J. Lecubin. D. Zevaco, and C. Blanpain (Institut Pythéas, Marseille, France).

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
