# Peer review of "Phytoplankton reaction to an intense storm in the north-western Mediterranean Sea"

_EGUsphere, 2022_

## Author Comment (AC2)

**Reviewer 1**

The study is well designed for capturing storm induced variation over the upper with in-situ observations from glider and cruise. Its impact is further quantified with atmospheric model and satellite observations. The center of storm is mostly captured along the glider track and the storm induced dynamics is clearly identified. Findings are generally persuading and interesting. A minor revision is suggested for addressing the following comments before the paper being accepted for publication.

The authors would like to thank you for your precious time, and positive and constructive comments. We have carefully addressed all the comments, questions and suggestions. All of them will be included in the revised version of the manuscript.
(Color legend : comments and questions in green, answers in blue, new text proposal in orange)

Major comments:

1. What is the spatial resolution and quality for the satellite observations? Though multiple algorithms are applied for the chlorophyll dataset and their results are highly consistent, the cloud coverage can be an issue for contaminating the observations. More details are needed to describe the measurements.

Yes, you will find below the information on all the products. We propose to put in the text only the ones used for the results (in orange).

. SSH and associated geostrophic currents

- "MEDITERRANEAN OCEAN GRIDDED L4 SEA SURFACE HEIGHTS AND DERIVED VARIABLES" (SEALEVEL_MED_PHY_L4_NRT_OBSERVATIONS_008_050, now SEALEVEL_EUR_PHY_L4_NRT_OBSERVATIONS_008_060, https://resources.marine.copernicus.eu/product-detail/SEALEVEL_EUR_PHY_L4_NRT_OBSERVATIONS_008_060/INFORMATION) : 0.125° x 0.125°, multi-satellite

. SST
- "MEDITERRANEAN SEA - HIGH RESOLUTION AND ULTRA HIGH RESOLUTION L3S SEA SURFACE TEMPERATURE (SST_MED_SST_L3S_NRT_OBSERVATIONS_010_012, https://resources.marine.copernicus.eu/product-detail/SST_MED_SST_L3S_NRT_OBSERVATIONS_010_012/INFORMATION) : 0.01° × 0.01°, strict temporal window (local nightime), to avoid diurnal cycle and cloud contamination. provides supercollated (merged multisensor, L3S) SST data remapped over the Mediterranean Sea.

- "MEDITERRANEAN SEA HIGH RESOLUTION AND ULTRA HIGH RESOLUTION SEA SURFACE TEMPERATURE ANALYSIS (SST_MED_SST_L4_NRT__OBSERVATIONS_010_004, https://resources.marine.copernicus.eu/product-detail/SST_MED_SST_L4_NRT_OBSERVATIONS_010_004/INFORMATION): 0.01° × 0.01°, nighttime images, multi-satellite

. Chl
-    "GLOBAL OCEAN CHLOROPHYLL FROM SATELLITE OBSERVATIONS" (OCEANCOLOUR_GLO_CHL_L3_NRT_OBSERVATIONS_009_032,                now

OCEANCOLOUR_GLO_BGC_L3_NRT_009_101,
https://resources.marine.copernicus.eu/product-detail/OCEANCOLOUR_GLO_BGC_L3_NRT_009_101/INFORMATION):
4km x 4km, ACRI-ST company,  multi-satellite, hereafter called Chl_ACRI

- "MEDITERRANEAN SEA SURFACE CHLOROPHYLL CONCENTRATION FROM MULTI SATELLITE OBSERVATIONS"
(OCEANCOLOUR_MED_CHL_L3_NRT_OBSERVATIONS_009_040,             now OCEANCOLOUR_MED_BGC_L3_NRT_009_141,
https://resources.marine.copernicus.eu/product-detail/OCEANCOLOUR_MED_BGC_L3_NRT_009_141/INFORMATION ):
1km x 1km, multi-satellite, hereafter called Chl_MEDOCL3

- "MEDITERRANEAN SEA DAILY INTERPOLATED SURFACE CHLOROPHYLL CONCENTRATION FROM MULTI SATELLITE OBSERVATIONS"
(OCEANCOLOUR_MED_CHL_L4_NRT_OBSERVATIONS_009_041,             now OCEANCOLOUR_MED_BGC_L4_NRT_009_142,
https://resources.marine.copernicus.eu/product-detail/OCEANCOLOUR_MED_BGC_L4_NRT_009_142/INFORMATION):
1km x 1km, multi-satellite, hereafter called Chl_MEDOCL4

Concerning the cloud coverage: yes, it can be an issue as the L4 Chl product, filling the cloud gaps with climatology, is not very accurate in our case, as can be seen in Fig.3a where the correlation between Chl_MEDOCL4 and in situ is worse than the one between Chl_MEDOCL3 and Chl_insitu. Nevertheless, the purposes of using satellite products are :
- having a synoptic view for cruise guidance (using SPASSO) and general hydrodynamic zones determination (Fig 4a)
- perform a qualitative comparison with Chl_insitu to assess the question of the Chl satellite performance with respect to data. The answer is that the qualitative performance is good in general, and follows the in situ trend,  but that the quantitative performance is not sufficient, especially as far as the storm effect is concerned (see Fig. 10b).

2. The storm induced variations are largely varying depending on the feature of the storm. For example, prominent changes are identified with storms with large intensity and slow moving (Wang, 2020). The frequency of storm and their associated intensities in the Mediterranean Sea should be described; thus, the readers have a better understanding for the representative of investigated storm.

We will add some more details to the paragraph L215-219, that will be moved to the discussion.

"The analysis of 30 years of coastal data in the South of France (Toulon) by Meteo France shows that winds of intensity $> 100\,\mathrm{km\,h^{-1}}$ occur on average 8 times per year, but only once every 4 years in May. Concerning winds of intensity $> 130\,\mathrm{km\,h^{-1}}$, they occur on average once every 2 years, and once every 30 years in May (http://tempetes.meteo.fr/spip.php?article221)"

→ "The analysis of 30 years of coastal data in the South of France (Toulon and Marignane) by Meteo France shows that the typical periods of wind occur at the end of winter and middle of autumn. In Toulon, winds of intensity > 100 km h$^{-1}$ occur on average 8 times per year, but only once every 4 years in May; winds of intensity > 130 km h$^{-1}$ occur on average once every 2 years, and once every 30 years in May. The total occurrences of different wind intensities for the whole 1981-2010 period are shown in Table 1 (http://tempetes.meteo.fr/spip.php?article221)."

| | | > 100 km h$^{-1}$ | > 110 km h$^{-1}$ | > 120 km h$^{-1}$ | > 130 km h$^{-1}$ |
|---|---|---|---|---|---|
| Toulon | Tot | 232 | 96 | 36 | 14 |
| | May | 8 | 1 | 1 | 1 |
| Marignane | Tot | 205 | 60 | 13 | 5 |
| | May | 6 | 0 | 0 | 0 |

Our episode of storm is rare both in intensity and period of the year, consequently not representative of the general climate until 2010 in the NW Mediterranean Sea. Nevertheless in the future important changes in both the frequency and the intensity of Mediterranean storms are expected (Lionello et al., 2006; Flaounas et al., 2021).

We will add in our discussion a reference to your suggested paper Wang, 2020 (see next comment).

3. The storm didn't necessary induce elevation in phytoplankton, especially in the stratified ocean with prominent subsurface chlorophyll maximum (Figure 13a). Similarly, there was no net increasing in chlorophyll resolved in the BGC-Argo observation in the northwest Pacific after a strong typhoon (Chai et al., 2021). The observed elevation in chlorophyll may be due to a redistribution, which should be further examined for different depth.

Yes, this is also our interpretation, thanks to the glider observation and Fig.13a and Fig.13b, that the increase in the surface Chl is most likely due to the dilution of the DCM, not an overall increase in depth. In addition, in our case the increase in nutrients seems to be linked to the uplift of the nitracline.

L300 : "However, the integrated fluorescence content in the upper 100 m did not show any noticeable variation (Fig. 13b). This indicates that the increase in chla concentration observed near the surface (Fig. 6) was likely due to the dilution by vertical mixing of the phytoplankton cells within the mixed layer."

Chai et al. 2021 says tropical and subtropical typhoons in open and deep waters do not always mix deep enough to allow the nutricline to reach surface waters, avoiding growth

enhancing while ocean color shows increase in chla concentration. A sentence using this paper and Wang 2020 will be included in the manuscript discussion:

"Typhoons can be compared to the type of storm observed in our study only by the intensity and duration of the winds triggering a fast decrease of surface temperature and an increase in surface chla. Most of the typhoons enhance chlorophyll surface concentration (Wang et al., 2020). Nevertheless, in open water tropical and sub-tropical areas, dilution phenomenon of the deep chlorophyll maximum after typhoons was warned to be source of overestimation of potential phytoplankton production when using only satellites observation, because the nitracline is not always affected (Chai et al., 2021). In our case, the deepening of the mixing due to the storm was accompanied by an increase in surface nutrients that could only be linked to the uplift of the nitracline, as we were far enough from coastal run-off influences. This mixing was related to the spreading and the increasing of the phytoplankton in the upper layer, leading to a possible dilution of the grazers favouring the pico-nanophytoplankton accumulation in the shallowing mixed lays a few days after (Morison et al., 2019). "

Minor comments:

1. The color shading for the boxes in Figure 3(a) is misleading. Please adjust to the same kind of color with different intensity.

Yes, we will change to shades of blue.

2. Ticks on the yaxis are misleading in Figure 11(b) since three curves with two axes. What is the meaning of the background shading?

It was indeed not straightforward to visualise that the green and red curves share the same axis and that is why we have put the title in green and the labels in red. A sentence will be added into the caption to make this point clearer: "The vertical axis tick colors indicate the associated curve. Similarly, ticks labels and titles written in two different colors indicate that two curves are associated with the same axis."

Similarly, the horizontal grid lines were misleading and will be removed. Finally, a sentence giving the meaning of the background shading meaning was added in the caption of Figure 11b: "Illustration of the newly-mixed waters (in cyan spans) and their direct surroundings (NC waters, in yellow spans)" → "Illustration of the newly-mixed waters (corresponding to the cyan background) and their direct surroundings (NC waters, corresponding to the yellow background)"

3. There are some inconsistencies in the formatting, like Line 252 the paragraph didn't finish.

Thank you, we will review the remaining inconsistencies in the formatting through the whole document.

4. Please modify the location where the figures to be embedded as many figures are inserted in the middle of a paragraph.

Yes, we will modify the figures location and avoid embedding them in the middle of paragraphs.

Chai, F., Wang, Y., Xing, X., Yan, Y., Xue, H., Wells, M., Boss, E. (2021), A limited effect of sub-tropical typhoons on phytoplankton dynamics. Biogeosciences, 18(3), 849-859.

Wang, Y. (2020), Composite of typhoon induced sea surface temperature and chlorophyll-a responses in the South China Sea, Journal Geophysical Research: Oceans, 125, e2020JC016243.

---

## Author Comment (AC3)

**Reviewer 2**

In this work, Barrillon et al. characterize the response of phytoplankton to a storm event in the NW Mediterranean Sea. In my opinion, this work is truly relevant. There are few studies out there that compare the before and after of phytoplankton response to short-term anomalous events that disrupt the ecosystem. Since there is a real possibility that such extreme events may become more frequent in the future, there is a dire need for more studies on this topic. Unfortunately, most of these works occur as a reaction to a given extreme event and, thus, lack a comprehensive methodology that may evaluate the impact it had, which is understandable. This is not the case of this manuscript, as Barrillon et al. clearly tried to use as much as they could to characterize this event: in-situ ship-based sampling before and after the storm, a glider sampling during the storm, as well as remote sensing and modelled data to complement the in-situ data. Therefore, this is an important work and a good example on how various sources of data should be integrated to study a short-term event.

While methodology is sound, the writing is overall good and its conclusions are relevant and supported by the results, I do have a few gripes with the manuscript that I believe should be resolved before being accepted. Therefore, for my part, I recommend major revisions.

We are very grateful to the reviewer for the careful reading and relevant comments about the manuscript. All the remarks have been addressed below and the associated modifications will be performed in the revised version of the manuscript.
(Color legend : comments and questions in green, answers in blue, new text proposal in orange)

I will now list my main questions or areas where I think the manuscript could be improved.

- In the introduction, the goals of the work are not explicitly stated. There is a large paragraph detailing the FUMSECK cruise, some overall methodology and its aims, but these are the cruise's aims, not this work's aims. Clearly stating the objectives and linking them with the methodology and results would help the reader navigating through the substantial number of results described in this work.

Yes, the new introduction will be as follows, also taking into account the other comments about introduction (changes in orange).

"Marine environments are subject to short-term events whose effects on biogeochemical processes can be substantial. This is the case of desert dust deposit on oligotrophic areas (Guieu et al., 2014), volcanic ash deposit (Hammes et al., 2010), submarines sources of iron (Guieu et al., 2018), and sudden mixing of the water column from typhoons (Wang et al., 2020). Even the classical phytoplankton spring bloom can vary in intensity and spatial extent

depending on the amount of previous short term storms (Ferreira et al., 2021). Effects on micro-organisms activity and diversity encompass sudden changes compared to the previous conditions. Indeed, depending on the redistribution of nutrients, the intensity of any turbulences, the light conditions and the mixing of different water masses, the phytoplankton community can either collapse or grow, affecting carbon export by generating decoupling phenomenon between production and remineralization (Henson et al., 2019).

Meteorological impulse wind events such as storms, and their effects on oceanic physics and even more on biogeochemistry, are poorly explored with in situ data. Such events generate mixing and stirring of the surface layer and can trigger transitional peaks in primary production, mainly explained by nitracline shoaling and grazers dilution (Lomas et al., 2009; Menkes et al., 2016). In oligotrophic ocean conditions, Babin et al. (2004) and Han et al. (2012) observed from satellite ocean colour the sudden and large increase in chlorophyll-a, lasting several weeks, after summer hurricane-storms. The resulting increase in surface chlorophyll-a (chla) reached values close to those from the spring bloom (Babin et al., 2004) with potential primary production comparable to the one induced by some mesoscale (~ 10 – 100 km horizontal range) eddies, but could not reach further processes understanding due to the lack of in situ observations. Only a few studies have combined high-resolution physical descriptions of wind events with a phytoplankton resolution at the functional group level. Some coastal studies, such as Fuchs et al. (2022), have evidenced pico-nanophytoplankton abundance and biomass rises for most phytoplankton groups, within two to four days following wind-induced events at a coastal station located in the north-western (NW) Mediterranean Sea in stratified conditions. Again, the authors showed that extreme events can generate daily biomass increases of the same order of magnitude as those observed during the spring bloom. Similarly, Anglès et al. (2015) studied the response of nano-microphytoplankton to tropical cyclones generating wind-physical forcing and substantial rains in the Western Gulf of Mexico. They highlighted strong increases in plankton abundance following the storms with delays consistent with Fuchs et al. (2022). These storms observed on either coastal Mediterranean systems or tropical open ocean may potentially exert a strong control on both primary production and community structure also in the Mediterranean open ocean, thus playing a potentially important biogeochemical role on the whole basin. However, in our knowledge no such event in the Mediterranean open sea has been reported in the past.

The classical spring bloom as observed in temperate oceans is triggered by the shoaling of the mixed layer when passing from the winter convection to the spring stratification (Behrenfeld, 2010), which ends when no more nutrients are available in the euphotic layer or when grazers overpass the phytoplankton growth capacity. This is particularly the case in the NW Mediterranean Sea characterised by winter deep convection (Houpert et al., 2016; Testor et al., 2018) and by spring blooms of different intensities that can be detected from satellite images (d'Ortenzio and Ribera d'Alcalà, 2009; Mayot et al., 2016). This area is affected by strong northerly winds, and their intensity in winter defines the bloom intensity (Conan et al., 2018). In summer stratified conditions, impulse wind events could induce submesoscale (~ 1 – 10 km horizontal range) vertical mixing and trigger patches of high phytoplankton production. Yet, observing the effect of these events on phytoplankton dynamics and distribution is challenging, especially during stratified oligotrophic conditions, and requires the deployment of dedicated automated and high-frequency sampling tools. Indeed, the mixing of the water column may bring microorganisms from deep to surface

layers, affects their physiological properties due to photoacclimation processes, and has an impact on carbon-to-chlorophyll ratios used to run primary production models at large scales (Sathyendranath et al., 2020). In addition, some scarce observations at the functional group level evidence daily adaptation processes rather than community changes after water column mixing (Thompson et al., 2018) or a taxonomical dependency in physiological strategies (Graff and Behrenfeld, 2018). Being able to monitor phytoplankton distribution at a functional level, by integrating small and rapid scale dynamics into larger space and time scales would precise the role of phytoplankton in biogeochemical processes.

The objective of this paper is to study in situ physical and biological effects of a particularly intense episode of wind in spring 2019 in the Ligurian Sea (NW Mediterranean Sea). The methodology is to combine data taken during the FUMSECK cruise (Facilities for Updating the Mediterranean Submesoscale - Ecosystem Coupling Knowledge, https://doi.org/10.17600/18001155, PI S. Barrillon (Barrillon et al., 2020)) when this episode happened. High-resolution physical properties, chla and phytoplankton were measured and combined to show abrupt changes in water characteristics, and phytoplankton abundances and physiological properties in surface waters."

- For a work in which its conclusions revolve around the "role of storms on the biogeochemistry and ecology of the Mediterranean open sea (…)", I saw very few references in the introduction to works focusing on other than phytoplankton abundance or biomass changes. For instance, the authors could have discussed other studies that have approached the potential impact of such short-term events on carbon export (e.g., Hamme et al., 2010; Henson et al., 2012, 2013; Ferreira et al., 2022). Regarding this matter, I would be curious to see some remote sensing POC images/data before and after the storm. These may even tie in nicely with the conclusions of the article, if they reveal something interesting.

We thank you for the suggested references. After reading the suggested papers, we propose to add in the introduction several examples to illustrate the impact of short events on biogeochemical processes (first paragraph of the new introduction):

"Marine environments are subject to short-term events whose effects on biogeochemical processes can be substantial. This is the case of desert dust deposit on oligotrophic areas (Guieu et al., 2014), volcanic ash deposit (Hammes et al., 2010), submarines sources of iron (Guieu et al., 2018), and sudden mixing of the water column from typhoons (Wang et al., 2020). Even the classical phytoplankton spring bloom can vary in intensity and spatial extent depending on the amount of previous short term storms (Ferreira et al., 2021). Effects on micro-organisms activity and diversity encompass sudden changes compared to the previous conditions. Indeed, depending on the redistribution of nutrients, the intensity of any turbulences, the light conditions and the mixing of different water masses, the phytoplankton community can either collapse or grow, affecting carbon export by generating decoupling phenomenon between production and remineralization (Henson et al., 2019)."

We also propose to include a sentence in the discussion:

"This could in turn foster the integrated primary production by enhancing phytoplankton division rate and biomass (Behrenfeld, 2010) which, when grazers are diluted, is related

to higher organic carbon export efficiency (Henson, 2019). This phenomenon was also observed after winter storms in the Sargasso Sea, where diatoms increase was maximal within two days after shoaling of the mixed layer depth (Krause, 2009). These pulsed production events could be responsible for up to 20% of the global primary production in the Sargasso Sea (Lomas et al., 2011)."

Regarding the POC analysis, we found this NASA product :
https://oceancolor.gsfc.nasa.gov/atbd/poc/#sec_6

A quick look seems to show some cloud issues during the storm (04, 05 of May). Nevertheless, looking at the region (images extracted from the web site, purple circle) on the 30th of April (before the storm) and the 06th of May (just after the storm), an increase of POC could be possible. This study would require some more time to deepen, we propose to perform this analysis when the revised manuscript will be submitted.

[Figure]

2019/04/30, extracted from https://oceancolor.gsfc.nasa.gov/showimages/MODISA/IMAGES/POC/L3/2019/0430/AQUA_MODIS.20190430.L3m.DAY.POC.poc.4km.nc.png

2019/05/06, extracted from https://oceancolor.gsfc.nasa.gov/showimages/MODISA/IMAGES/POC/L3/2019/0506/AQUA_MODIS.20190506.L3m.DAY.POC.poc.4km.nc.png

- Some paragraphs or portions of the manuscript are a bit verbose and could be shortened or even removed. For instance, in lines 53-64, there is an exhaustive description of the FUMSECK cruise. This description could be shortened and most of it could be integrated into the Material and Methods section to avoid redundancy.

Yes, we will remove the paragraph L53-64 from the introduction, shorten and embed it in the material and methods introduction as follows :

"The FUMSECK cruise aimed at combining physical and biological oceanography for the study of mesoscale and submesoscale dynamics, which imply structures such as eddies, filaments, or fronts over a horizontal spatial range of 1 to 100 km, a vertical one of 0.1 to 1 km, and a temporal range of days to a few weeks (Giordani et al., 2006; Ferrari and Wunsch, 2009; McWilliams, 2019). It took place from 30 April 2019 to 7 May 2019, in the Ligurian Sea (NW Mediterranean Sea), onboard the RV Téthys II. During this cruise, we deployed towed instruments and an underwater glider (Testor et al., 2019) to measure physical properties at high resolution. These measurements have been paired with shipboard measurements of

phytoplankton functional groups from an automated pulse-shape recording flow cytometer, based on cell sizes and pigment contents (Dugenne et al., 2014; Thyssen et al., 2014; Bonato et al., 2015; Louchart et al., 2020). Figure 1 shows the cruise and the glider trajectories. Right after the storm for which we had to take shelter, the ship came back to the wind-exposed zone to collect data. Meanwhile, the glider stayed in the storm-exposed zone all along and collected data. In addition to in situ data, we exploited satellite data to guide the cruise and obtain a synoptic view of the region and a meteorological model to study the storm."

We will also remove the mention to the stations, and the following paragraph :

"Several in situ instruments for measuring physics and biogeochemistry were deployed and are described in this section within the first two parts: transect measurements, and glider. The satellite data exploited to guide the cruise and obtain a synoptic view of the region are described in the third part, followed by the meteorological model. The last part deals with the comparison of the fluorescence and chla concentrations from the different measurements."

Also, passages such as 66-69 and 73-76 are redundant. There is no need to state what the results section will show after the material and methods because that will become clear for the readers as they continue to read the manuscript. I recommend looking at such situations across the manuscript to keep the text as straightforward as possible for the reader.

Yes, such passages will be removed from the introduction, and the material and methods introduction.

- I think the sampling scheme could be clearer. For instance, there are underway surface water measurements of ADCP, SSS, SST and chl-a via fluorometer throughout the entire cruise (30/04/2019-07/05/2019). Then there is also an MVP which was deployed along seven different transects (only two are shown, as far as I noticed) and again sampled temperature, salinity, chl-a via fluorometer. It also included a plankton counter. Yet we only know the timing and duration of transect 1 (30/April) and 7 (5-6/May). Figure 6 is exhibiting the transects and its measured variables, but this should be clearly stated. Furthermore, while I was reading the manuscript, I was frequently unsure if what is being shown is the temperature/salinity data from the MVP or from the underway system. Inorganic nutrients were also sampled at 26 locations (or, at least, 26 samples were collected) and chl-a again (now via laboratory fluorometer) was sampled at 20 locations, all at surface. Judging from Figure 7, I think most of these chl-a and nutrients samples match, but, again, it should not be necessary for the reader to carefully compare figures and count stations to understanding the overall picture of what was done and how. What are the seven stations presented in Figure 1? Did multiple-variable sampling occur in these stations or are they just the location where the ship turned and began a new transect? For instance, in Figure 7, does the discrete in-situ sampling stations match the seven stations in Figure 1? Overall, I suggest revisiting the methodology section. One idea that may help could be including a table that lists all variables sampled, abbreviation and the source (ship underway, MVP, glider, discrete).

To answer your comment we propose several actions for the revised manuscript:

- The stations correspond to vertical velocity measurements, which are not exploited in this paper → we propose to remove them from the figures and the text.
- Add two zoomed plots as Fig1b and Fig1c, one showing the MVP transects and the other one showing the in situ samplings locations (one color for the common Chl/nutrients stations, another color for the stations not in commun)
- Remove from the material and methods the measurements not used in the paper
- Add the table below in the text, following your suggestions.

| Observables | Abbreviation/ name | Vertical Range | Sampling | Source |
|---|---|---|---|---|
| Horizontal currents | ADCP currents | 18 - 562 m | all cruise, 0.4 km resolution | VM-ADCP |
| | geostrophic currents | first meters | daily, 2 April to 3 July 2019 | Satellite |
| Connservative Temperature | $\Theta$_tsg | 2m | all cruise, 0.2 km resolution | TSG |
| | $\Theta$_mvp | 0-308 m | 7 transects, 1.3 km resolution | MVP |
| | $\Theta$_glider | 0-600 m | 2 transects, 1 km resolution | Glider |
| Absolute Salinity | $S_A$_tsg | 2 m | all cruise, 0.2 km resolution | TSG |
| | $S_A$_mvp | 0-308 m | 7 transects, 1.3 km resolution | MVP |
| | $S_A$_glider | 0-600 m | 2 transects, 1km resolution | Glider |
| Fluorescence | RFluo_tsg | 2 m | all cruise, 0.2 km resolution | TSG |
| | RFluo_cyto | 2 m | 400 samples, 3.9 km resolution | AFCM |
| | FL_npq | 0-600 m | 2 transects, 1 km resolution | Glider |
| Chlorophyll-a | Chl_tsg (converted) | 2 m | all cruise, 0.2 km resolution | TSG |
| | Chl_insitu | 2 m | 20 samples | in situ |
| | Chl_cyto (converted) | 2 m | 400 samples, 3.9 km resolution | AFCM |
| | Chl_ACRI Chl_MEDOCL3 Chl_MEDOCL4 | first meters | daily, 2 April to 3 July 2019 | Satellite |
| Nutrients | Phosphate ($PO_4^{3-}$) Nitrate ($NO_3^-$)  Nitrite ($NO_2^-$)  Silicate ($Si(OH)_4$) | 2 m | 26 samples | in situ |
| Phytoplankton | Abondance, size, | 2 m | 400 samples, 3.9 km | AFCM |

| observables | biovolume, biomass | | resolution | |
|---|---|---|---|---|

- Finally, I think the discussion can be improved since it seems slightly superficial. For a large body of results (pages 9-21, including figures), a ~1 page discussion is quite short, particularly when the results are good. I feel the discussion lacks a comparison to other works on storm events, both in the Mediterranean and other areas. The authors do briefly compare some results with the OSCAHR cruise, yet this cruise occurred in November and did not sample a storm event (as far as its mentioned in the manuscript). Therefore, why would the results be directly comparable? This is not to say that this comparison is not valuable, but a better contextualization should be included.

We agree that we have to deepen the discussion. We will add a comparison with the datasets collected by Boudgriga et al., 2022, crossing the area during a similar period of the year (see the modified discussion at the end of the major comments below), and Latasa et al., 2022, showing similar trends in the Western Mediterranean Sea.

- Also, some conclusions within the discussion feel rushed and could do with better contextualization and arguments. For instance, in lines 317-318, the authors state that 'This suggests that cells did not have time to photo-acclimate or that different species were involved" after comparing the ratio of chl-a between chl-a in "cold" and "warm" waters. First, this information is not enough to make these statements. Secondly, this is the only mention of photoacclimation in the manuscript, except for line 46 in the introduction. Finally, this conclusion is quickly forgotten since the paragraph moves on and compares the increase in chl-a with a previous work from 2000.

We agree this part is missing information. A change is proposed in the new discussion.

- Again, in lines 333-334, the authors now suggest the drop in carbon/chl-a ratio is a "clear signature of a sudden change in phytoplankton cell physiology and translated the unadapted configuration of the cells to high light condition". Why is it a clear signature? Why is one thing related to another? It is up to the authors to make the 'bridge' between the results and the conclusions, not the reader. Moreover, the paragraph ends with this sentence, without any comparison to other studies or without a discussion of its implications.

In accordance with the previous comment, changes are suggested in the text in order to make the reading more friendly.

New discussion:

[revised manuscript text omitted]

Typhoons can be compared to the type of storm observed in our study only by the intensity and duration of the winds triggering a fast decrease of surface temperature and an increase in surface chla. Most of the typhoons enhance chlorophyll surface concentration (Wang et al., 2020). Nevertheless, in open water tropical and sub-tropical areas, dilution phenomenon of the deep chlorophyll maximum after typhoons was warned to be source of overestimation of potential phytoplankton production when using only satellites observation, because the nitracline is not always affected (Chai et al., 2021).

In our case, although the increase in chla after the deepening of the mixed layer depth during post-bloom periods and linked to wind events is not obvious as demonstrated by Andersen and Prieur (2000), the deepening of the mixing due to the storm was accompanied by an increase in surface nutrients that could only be linked to the uplift of the nitracline, as we were far enough from coastal run-off influences. This mixing was related to the spreading and the increasing of the phytoplankton in the upper layer, leading to a possible dilution of the grazers favouring the pico-nanophytoplankton accumulation in the shallowing mixed lays a few days after (Morison et al., 2019). This could in turn foster the integrated primary production by enhancing phytoplankton division rate and biomass (Behrenfeld, 2010) which, when grazers are diluted, is related to higher organic carbon export efficiency (Henson, 2019). This phenomenon was also observed after winter storms in the Sargasso Sea, where diatoms increase was maximal within two days after shoaling of the mixed layer depth (Krause, 2009). These pulsed production events could be responsible for up to 20% of the global primary production in the Sargasso Sea (Lomass et al., 2011).

Our observations captured the short-term physical and phytoplankton response to a storm, with rapid and strong changes observed but without the possibility to follow in situ post-conditions. Although not representative of what happens in the entire mixed water

column, satellite data showed an effect on surface temperature and chla within the ship-glider storm geographical zone (longitudes between 8° E and 8° 30' E and latitudes between 43° 30' N and 43° 42' N). In this zone the mean value of SST was lower during four days after the storm (6-10 May, 14.8ºC) than between the 20 April-20 May period (15.1° C), while the mean value of Chl_ACRI was higher (0.44 ng mL$^{-1}$ with respect to 0.32 ng mL$^{-1}$), suggesting the pico-nanophytoplankton size classes could have had the time to grow and accumulate, as their growth rate is close to one to two divisions a day when nutrient and light are available (Morison et al., 2019).

For future work, the objective will be to study the medium to long-term response, after the so-called reaction period, and for each observed phytoplankton group. Indeed, such events are critical, as they may affect the primary production annual budgets.

**Minor comments (lines on the left):**

3: Please remove or change 'violent' for a more adequate term (e.g., intense).

Yes, intense.

4: NW is written as 'north-western', yet the title includes 'northwestern'. Uniformize.

Yes, we will uniformize all over the text.

8: missing of: factor of two

Yes.

9: missing of: and of seven

Yes.

24: missing have: have combined

Yes (see new introduction).

26: what does 'have evidenced pico-nanophytoplankton abundance and biomass responses' mean? Did it increase, lower?

Yes, an increase for most of the groups (see new introduction).

29: remove have: 'have studied'

Yes.

34: Are you suggesting that no previous cases of storms shaping primary production and phytoplankton community structure have been reported? It is not clear if this only refers to the NW Mediterranean, the entire Mediterranean or if it also includes other systems.

It refers to the Mediterranean open sea, changed in the new introduction.

38: missing the: overpass the phytoplankton growth capacity

Yes.

38: you already have north-western written in line 27, you can already use NW

Yes.

41: This area

Yes.

45: Add 'may': 'the mixing of the water column may bring microorganisms from deep to surface layers and affect their photophysiological properties (…)'

Yes.

62-64: methods?

Yes, this paragraph will be moved to Material and Methods.

91: were performed

Yes.

95 and 102: please specify that these are surface-only samples

Yes.

Figure 2 caption: Orgnano and Unidentified particles groups have the same colour (green dots). Use light and dark green, for instance, to differentiate them in the caption.

The Orgnano and Unidentified particles already had dark and light green colors, respectively, but as the Orgnano are rare and in the top-right part of the cytogram (Fig.2a), they are not visible enough. We propose to change the color of Orgnano to fushia.

149-150: It should have been calibrated prior to the cruise. Nevertheless, how good is the agreement with ship-based chl-a? Since the glider is the only source of data during the storm, this should be presented as supplementary material or, at least, the R, p-val, error and N should be indicated in the text.

No pre-cruise calibration of the Ecopuck was carried out. Nevertheless, we observe a good statistical agreement between the measurements of the glider with those taken from the ship (see Figure). Over a sample of N = 33 glider profiles where the glider-ship distance is lower than 40km, surface chla fluorescence from the ship and 0-10m average from the glider have a correlation coefficient of R = 0.76  (with a significant p-value of 2.5e-7) and a mean standard error of 0.067 ng mL$^{-1}$, well below of the amplitude of the observed signal during the storm (approx. 0.5 ng mL$^{-1}$). Values from the onset of the storm have been excluded (grey values after 05 May) since the glider was experiencing different conditions than the ship sheltering from the bad weather. At the end of the time series, when the glider was recovered, the values between the two platforms agree well again, which gives us a good confidence in the chla fluorescence signals described by the glider's sensor during its mission.

We could put this paragraph and the associated figure in supplementary material.

[Figure]

156: swap SSH and sea surface height

Yes.

157: swap SST and sea surface temperature

Yes.

157: there is no such thing as sea surface chl-a. Satellite chl-a does not capture only surface chl-a.

Right, we will replace sea surface chla by chla integrated over the first few meters, when satellite chla is concerned.

159-160: please provide a bit more detail on the satellite products instead of just referring to another paper. You may leave the citation, but please add a brief description, just mentioning the name of the products or sensors and their resolution.

Yes, you will find below the information on all the products. We propose to put in the text only the ones used for the results (in orange).

. SSH and associated geostrophic currents

- "MEDITERRANEAN OCEAN GRIDDED L4 SEA SURFACE HEIGHTS AND DERIVED VARIABLES" (SEALEVEL_MED_PHY_L4_NRT_OBSERVATIONS_008_050, now SEALEVEL_EUR_PHY_L4_NRT_OBSERVATIONS_008_060, https://resources.marine.copernicus.eu/product-detail/SEALEVEL_EUR_PHY_L4_NRT_OBSERVATIONS_008_060/INFORMATION) : 0.125° x 0.125°, multi-satellite

. SST
- "MEDITERRANEAN SEA - HIGH RESOLUTION AND ULTRA HIGH RESOLUTION L3S SEA SURFACE TEMPERATURE (SST_MED_SST_L3S_NRT_OBSERVATIONS_010_012, https://resources.marine.copernicus.eu/product-detail/SST_MED_SST_L3S_NRT_OBSERVATIONS_010_012/INFORMATION) : 0.01° × 0.01°, strict temporal window (local nightime), to avoid diurnal cycle and cloud contamination. provides supercollated (merged multisensor, L3S) SST data remapped over the Mediterranean Sea

- "MEDITERRANEAN SEA HIGH RESOLUTION AND ULTRA HIGH RESOLUTION SEA SUR- FACE TEMPERATURE ANALYSIS (SST_MED_SST_L4_NRT_ _OBSERVATIONS_010_004,https://resources.marine.copernicus.eu/product-detail/SST_MED_SST_L4_NRT_OBSERVATIONS_010_004/INFORMATION): 0.01° × 0.01°, nighttime images, multi-satellite

. Chl
- "GLOBAL OCEAN CHLOROPHYLL FROM SATELLITE OBSERVATIONS" (OCEANCOLOUR_GLO_CHL_L3_NRT_OBSERVATIONS_009_032, now OCEANCOLOUR_GLO_BGC_L3_NRT_009_101, https://resources.marine.copernicus.eu/product-detail/OCEANCOLOUR_GLO_BGC_L3_NRT_009_101/INFORMATION) : 4km x 4km, ACRI-ST company, multi-satellite, hereafter called Chl_ACRI

- "MEDITERRANEAN SEA SURFACE CHLOROPHYLL CONCENTRATION FROM MULTI SATELLITE OBSERVATIONS" (OCEANCOLOUR_MED_CHL_L3_NRT_OBSERVATIONS_009_040, now OCEANCOLOUR_MED_BGC_L3_NRT_009_141, https://resources.marine.copernicus.eu/product-detail/OCEANCOLOUR_MED_BGC_L3_NRT_009_141/INFORMATION ): 1km x 1km, multi-satellite, hereafter called Chl_MEDOCL3

- "MEDITERRANEAN SEA DAILY INTERPOLATED SURFACE CHLOROPHYLL CONCENTRATION FROM MULTI SATELLITE OBSERVATIONS" (OCEANCOLOUR_MED_CHL_L4_NRT_OBSERVATIONS_009_041, now OCEANCOLOUR_MED_BGC_L4_NRT_009_142,

https://resources.marine.copernicus.eu/product-detail/OCEANCOLOUR_MED_BGC_L4_NRT_009_142/INFORMATION):
1km x 1km, multi-satellite, hereafter called Chl_MEDOCL4

167: reference for the ECMWF model?

We will add these two references :

- Bechtold, P., R. Forbes, I. Sandu, S. Lang, and M. Ahlgrimm, 2020: A major moist physics upgrade for the IFS. ECMWF Newsletter, No. 164, ECMWF, Reading, United Kingdom, 24–32, https://www.ecmwf.int/node/19720.

- Ben Bouallègue, Z., 2020: Accounting for representativeness in the verification of ensemble forecasts. ECMWF Tech. Memo. 865, ECMWF, 28 pp., https://www.ecmwf.int/node/19544.

174: techniques instead of sources

Yes.

179: in this context, this R2 could be higher.

We think we had misqualified the names in the equation. We have changed the Chl_tsg and Chl_cyto by the Chl_insitu.

→ "Fluorescence from the TSG (RFluo_tsg) was converted into units of chla concentration (Chl_tsg, ng mL$^{-1}$) using the significant correlation with Chl_insitu, Chl_insitu = 0.85 x RFuo_tsg - 0,19, r2 = 0.79, n = 20.

AFCM chla concentration (Chl_cyto) was estimated from the Rfluo_cyto. Values were normalised with 2 µm Polyscience beads, and multiplied by the abundance of each group to get the total normalised Rfluo_cyto per unit of volume (nRFluo_cyto (a.u mL$^{-1}$)). nRFluo_cyto was then compared to the Chl_insitu (Fig. 3a and b). A set of samples from a minicosm experiment (PIANO, unpublished data), acquired with the same chla extraction protocol and the same Cytosense instrument was added to the observations. These samples presented higher chla concentration values, strengthening the relationship. The linear relation between nRfluo_cyto and Chl_insitu was used to estimate chla concentration for each AFCM phytoplankton group (Chl_cyto, ng mL$^{-1}$) following the linear regression Chl_insitu = 0.11 x nRFluo_Cyto, r2 = 0.97, n = 41 (Fig. 3b). The origin of the linear regression was not significantly different from zero."

In the case this was not the reviewer's request, we found the correlation between chla from samples analysed in the lab and fluorescence from a fluorimeter not so bad in our case. Indeed, as a comparison:

Marrec et al., 2016 : $r^2$=0.50, n=41, chla varying from 0.08 to 0.41 µg $L^{-1}$

Thyssen et al., 2015 : $r^2$=0.86, n=12, chla varying from 0.21 to 7.80 µg $L^{-1}$

Our study : $r^2$ = 0.79, n=20, chla varying from 0.07 and 0.82 µg $L^{-1}$

189: again, remove sea surface.

Yes.

191-192: the comparison period should actually be much shorter since the main ocean colour sensors overpass occurs between 10h-13:30h, depending on the sensor (see section 3.1 in Sathyendranath et al., 2019; Remote Sensing, 19(19), 4285). I would try rerunning the comparisons with a shorter period, it is possible the results may improve.

We agree that the 10h-13h30 period is indeed more appropriate to perform the comparison: most of the correlations found with this time period increase with respect to the 6am-6pm time period. Yet, using this time period, the correlation between Chl_insitu and Chl_ACRI is performed on 4 points only, between Chl_insitu and MEDOC_L3 on 2 points and between Chl_insitu and Chl_MEDOCL4 on 5 points, which is far too low for this time period to be used in practice.

Figure 3:

- how does the R between MEDOCL3 and MEDOCL4 is equal to 1, but the R between MEDOCL3 and Chl_insitu is 0.84 and MEDOCL4 and Chl_insitu is 0.65?

MEDOCL3 has gaps due to cloud coverage, and MEDOCL4 fills the gap with some climatology. Thus, L4 points are composed of the L3 points plus some additional climatology-based interpolated points. The correlation between L3 and L4 is therefore performed only on the "L3 points": by construction the correlation is 1. Similarly, the Chl_insitu/MEDOC_L3 correlation is performed on 10 points whereas the Chl_insitu/MEDOC_L4 correlation is performed on 17 points (10 "L3 points" and 7 interpolated points). The worst correlation for Chl_insitu/MEDOC_L4 shows that the climatology interpolation does not match our in situ observations here.

- Where does the N=4555 come from when comparing satellite and in-situ data? Satellite data should be, at most, daily data unless the authors are working with geostationary sensors

We agree that this number can be misleading. Satellite data are indeed daily provided, on a lat/lon grid. We performed the association between each Chl_tsg data and the Chl satellite data on the same day and for the closest lat/lon pixel, then selected the ones where the Chl_tsg data is between 6:00-18:00 UTC day time. As a matter of fact, checking this, we found that the number 4555 in the text was not correct: Chl_ACRI n = 3522, Chl_MEDOCL3 n = 2094, Chl_MEDOCL4 n = 4498. We will correct these numbers.

- The colour palette for the correlation plot should be changed to a more uniform one (e.g., R=0 white, R=1 dark red)

Yes, we will change to shades of blue.

197-199: these are not results

Yes, we will move these lines to the Material and Methods.

201: why did you opt for MEDOCL4 when the relationship between satellite and in-situ was much better for MEDOCL3?

For this figure, the purpose was to define the mean dynamic zones during the cruise. We decided to use MEDOCL4 even if the correlation with in situ Chl is worse than the MEDOCL3 one,  to avoid the clouds that can create artificial features when averaging on several days.

204: I recommend changing the Chl-a units from ng/mL to either ug/L or mg/m3 since these options are more commonly used.

We understand your remark, as chlorophyll-a concentration is often written in µg/L. Our manuscript uses volumes a lot, and we are presenting all the data in units/mL to homogenise with the flow cytometry datasets. Indeed, if we would use L, we would be required to add a $10^3$ for each abundances presented in the tables and in the figures.

212: I recommend using m/s for wind speed. Also, the same units should always be used throughout the text (see line 222).

Yes, we will use m/s and homogenise through the text.

212: are these average or maximum intensities? Not clear.

They are the ranges of the intensities.

215-218: Again, these are not results from this work, unless you include them as supplementary material. Thus, this comparison would be more suitable in the discussion.

Yes (see new discussion).

224: The final sentence of the paragraph can be removed.

Yes.

237: rose instead of rised up

Yes.

299-300: this should also be in the discussion.

Yes.

302: the water column was

We mean general characteristics, we propose a rephrasing :

"At the time of the FUMSECK cruise, in May, the water column is generally well stratified with nearly undetectable surface nutrient availability (Pasqueron De Fommervault et al., 2015"

→ "In the NW Mediterranean Sea, in May, the water column is generally well stratified with nearly undetectable surface nutrient availability (Pasqueron De Fommervault et al., 2015, This was indeed the oceanographic setting for the FUMSECK cruise, before an intense storm dominated by north-westerly winds impacted the water column."

311-312: add percentages or values when comparing

Yes (see new discussion).

---

## Author Response (AR1)

**Reviewer 1**

The study is well designed for capturing storm induced variation over the upper with in-situ observations from glider and cruise. Its impact is further quantified with atmospheric model and satellite observations. The center of storm is mostly captured along the glider track and the storm induced dynamics is clearly identified. Findings are generally persuading and interesting. A minor revision is suggested for addressing the following comments before the paper being accepted for publication.

The authors would like to thank you for your precious time, and positive and constructive comments. We have carefully addressed all the comments, questions and suggestions. All of them are included in the revised version of the manuscript.

(Color legend: comments and questions in green, answers in blue, new text in orange)

Major comments:

1. What is the spatial resolution and quality for the satellite observations? Though multiple algorithms are applied for the chlorophyll dataset and their results are highly consistent, the cloud coverage can be an issue for contaminating the observations. More details are needed to describe the measurements.

Yes, you will find below the information on all the products, now in the paper in Appendix B.

- SSH and associated geostrophic currents

. "Mediterranean ocean gridded L4 Sea Surface Heights and derived variables" (SEALEVEL\_MED\_PHY\_L4\_NRT\_OBSERVATIONS\_008\_050, now SEALEVEL\_EUR\_PHY\_L4\_NRT\_OBSERVATIONS\_008\_060, https://resources.marine.copernicus.eu/product-detail/SEALEVEL\_EUR\_PHY\_L4\_NRT\_OBSERVATIONS\_008\_060/INE ORMATION): 0.125° x 0.125°, multi-satellite

- SST

- . "Mediterranean Sea High resolution and ultra high resolution L3S Sea Surface Temperature" (SST\_MED\_SST\_L3S\_NRT\_OBSERVATIONS\_010\_012, <a href="https://resources.marine.copernicus.eu/product-detail/SST\_MED\_SST\_L3S\_NRT\_OBSERVATIONS\_010\_012/INFORMA\_TION">https://resources.marine.copernicus.eu/product-detail/SST\_MED\_SST\_L3S\_NRT\_OBSERVATIONS\_010\_012/INFORMA\_TION</a>) :  $0.01^{\circ} \times 0.01^{\circ}$ , strict temporal window (local nighttime), to avoid diurnal cycle and cloud contamination. provides supercollated (merged multisensor, L3S) SST data remapped over the Mediterranean Sea.
- . "Mediterranean Sea high resolution and ultra high resolution Sea Surface Temperature analysis" (SST\_MED\_SST\_L4\_NRT\_OBSERVATIONS\_010\_004, https://resources.marine.copernicus.eu/product-detail/SST\_MED\_SST\_L4\_NRT\_OBSERVATIONS\_010\_004/INFORMAT ION): 0.01° × 0.01°, nighttime images, multi-satellite

- Chla

. "Global ocean Chlorophyll from satellite observations" (OCEANCOLOUR\_GLO\_CHL\_L3\_NRT\_OBSERVATIONS\_009\_032, now OCEANCOLOUR\_GLO\_BGC\_L3\_NRT\_009\_101, https://resources.marine.copernicus.eu/product-detail/OCEANCOLOUR\_GLO\_BGC\_L3\_NRT\_009\_101/INFORMATION): 4km x 4km, ACRI-ST company, multi-satellite, hereafter called ChI\_ACRI

- . "Mediterranean Sea surface Chlorophyll concentration from multi satellite observations" (OCEANCOLOUR\_MED\_CHL\_L3\_NRT\_OBSERVATIONS\_009\_040, now OCEANCOLOUR\_MED\_BGC\_L3\_NRT\_009\_141,
- https://resources.marine.copernicus.eu/product-detail/OCEANCOLOUR MED BGC L3 NRT 009 141/INFORMATION) : 1km x 1km, multi-satellite, hereafter called ChI\_MEDOCL3
- . "Mediterranean Sea daily interpolated surface Chlorophyll concentration from multi satellite observations"

(OCEANCOLOUR\_MED\_CHL\_L4\_NRT\_OBSERVATIONS\_009\_041, now OCEANCOLOUR\_MED\_BGC\_L4\_NRT\_009\_142,

https://resources.marine.copernicus.eu/product-detail/OCEANCOLOUR MED BGC L4 NRT 009 142/INFORMATION) : 1km x 1km, multi-satellite, hereafter called ChI\_MEDOCL4

Concerning the cloud coverage: yes, it can be an issue as the L4 ChI product, filling the cloud gaps with climatology, is not very accurate in our case, as can be seen in Fig.3a where the correlation between ChI\_MEDOCL4 and in situ is worse than the one between ChI\_MEDOCL3 and ChI\_insitu. Nevertheless, the purposes of using satellite products are :

- having a synoptic view for cruise guidance (using SPASSO) and general hydrodynamic zones determination (Fig. 4a)

- perform a qualitative comparison with Chl\_insitu to assess the question of the Chl satellite performance with respect to data. The answer is that the qualitative performance is good in general, and follows the in situ trend, but that the quantitative performance is not sufficient, especially as far as the storm effect is concerned (see Fig. 10b).

2. The storm induced variations are largely varying depending on the feature of the storm. For example, prominent changes are identified with storms with large intensity and slow moving (Wang, 2020). The frequency of storm and their associated intensities in the Mediterranean Sea should be described; thus, the readers have a better understanding for the representative of investigated storm.

We have added some more details, a reference to your suggested paper Wang (2020), and moved this part into the discussion.

"The analysis of 30 years of coastal data in the South of France (Toulon) by Meteo France shows that winds of intensity  $> 100 \text{ km h}^{-1}$  occur on average 8 times per year, but only once every 4 years in May. Concerning winds of intensity  $> 130 \text{ km h}^{-1}$ , they occur on average once every 2 years, and once every 30 years in May (http://tempetes.meteo.fr/spip.php?article221)"

→ "The analysis of 30 years of coastal data in the South of France (Toulon and Marignane) by Meteo France shows that the typical periods of intense wind occur at the end of winter and middle of autumn. In Toulon, winds of intensity > 27.8 m s-1 occur on average 8 times per year, but only once every 4 years in May; winds of intensity > 36.1 m s-1 occur on average once every 2 years, and once every 30 years in May. The total occurrences of different wind intensities for the whole 1981-2010 period are shown in Table 3. The wind

intensity of the studied storm, reaching a maximum of 36.1 m s-1, was rare in the Mediterranean Sea, and similar to the average wind intensity of the typhoons studied by Wang (2020).

| Location  | Wind intensity | $> 27.8{\rm ms^{-1}} \\ (> 100{\rm kmh^{-1}})$ | $> 30.6 \mathrm{ms^{-1}}$
(> 110 km h -1 ) | $> 33.3 \mathrm{ms^{-1}}$
(> 120 km h -1 ) | $> 36.1 \mathrm{ms^{-1}}$
(> 130 km h -1 ) |
|-----------|----------------|------------------------------------------------|----------------------------------------------------------|----------------------------------------------------------|----------------------------------------------------------|
| Toulon    | Tot            | 232                                            | 96                                                       | 36                                                       | 14                                                       |
|           | May            | 8                                              | 1                                                        | 1                                                        | 1                                                        |
| Marignane | Tot            | 205                                            | 60                                                       | 13                                                       | 5                                                        |
|           | May            | 6                                              | 0                                                        | 0                                                        | 0                                                        |

Table 3. Occurences of different wind intensities events for the 1981-2010 period in the South of France, in total and in May only (http://tempetes.meteo.fr/spip.php?article221).

Our episode of storm is rare both in intensity and period of the year, consequently not representative of the general climate until 2010 in the NW Mediterranean Sea. Nevertheless in the future important changes in both the frequency and the intensity of Mediterranean storms are expected (Lionello et al., 2006; Flaounas et al., 2021).

3. The storm didn't necessary induce elevation in phytoplankton, especially in the stratified ocean with prominent subsurface chlorophyll maximum (Figure 13a). Similarly, there was no net increasing in chlorophyll resolved in the BGC-Argo observation in the northwest Pacific after a strong typhoon (Chai et al., 2021). The observed elevation in chlorophyll may be due to a redistribution, which should be further examined for different depth.

Yes, this is also our interpretation, thanks to the glider observation and Fig.13a and Fig.13b, that the increase in the surface Chl is most likely due to the dilution of the DCM, not an overall increase in depth. In addition, in our case the increase in nutrients seems to be linked to the uplift of the nitracline.

Chai et al. (2021) says tropical and subtropical typhoons in open and deep waters do not always mix deep enough to allow the nutricline to reach surface waters, avoiding growth enhancing while ocean color shows increase in chla concentration. A paragraph using this paper and Wang (2020) have been included in the manuscript discussion:

"Typhoons can be compared to the type of storm observed in our study by the intensity and duration of the winds triggering a fast decrease of surface temperature and an increase in surface chla. Most of the typhoons enhance chlorophyll surface concentration (Wang et al., 2020). In open water tropical and sub-tropical areas, the dilution phenomenon of the deep chlorophyll maximum after typhoons was warned to be source of overestimation of potential phytoplankton production when using only satellite observation, because the nitracline is not always affected (Chai et al., 2021), and because chla from the deep chlorophyll maximum is not always related to higher biomass and production (Marañón et al., 2021). The increase in chla after the deepening of the mixed layer depth in post-bloom periods entailed by wind events is not obvious as demonstrated by Andersen and Prieur (2000). In our case, the

deepening of the mixing due to the storm was accompanied by an increase in surface nutrients that could only be linked to the uplift of the nitracline, as we were far enough from coastal run-off influences. This mixing was related to the spreading and the increasing of the phytoplankton in the upper layer in terms of biomass and chla. In addition, the mixing could lead to a possible dilution of the grazers favouring the pico-nanophytoplankton accumulation in the shallowing mixed layers a few days after (Morison et al., 2019). This could in turn foster the integrated primary production by enhancing phytoplankton division rate and biomass (Behrenfeld, 2010) which, when grazers are diluted, is related to higher organic carbon export efficiency (Henson et al., 2019). This phenomenon was also observed after winter storms in the Sargasso Sea, where diatoms increase was maximal within two days after shoaling of the mixed layer depth (Krause et al., 2009). These pulsed production events could be responsible for up to 20\% of the global primary production in the Sargasso Sea (Lomas et al., 2009)."

Minor comments:

1. The color shading for the boxes in Figure 3(a) is misleading. Please adjust to the same kind of color with different intensity.

Yes, we have changed to shades of blue.

2. Ticks on the yaxis are misleading in Figure 11(b) since three curves with two axes. What is the meaning of the background shading?

It was indeed not straightforward to visualise that the green and red curves share the same axis and that is why we have put the title in green and the labels in red. A sentence is added into the caption to make this point clearer: "The vertical axis tick colors indicate the associated curve. Similarly, ticks labels and titles written in two different colors indicate that two curves are associated with the same axis."

Similarly, the horizontal grid lines were misleading and are now removed. Finally, a sentence giving the meaning of the background shading is added in the caption of Figure 11b: "Illustration of the newly-mixed waters (in cyan spans) and their direct surroundings (NC waters, in yellow spans)"  $\rightarrow$  "Illustration of the newly-mixed waters (corresponding to the cyan background) and their direct surroundings (NC waters, corresponding to the yellow background)"

3. There are some inconsistencies in the formatting, like Line 252 the paragraph didn't finish.

Thank you, we have reviewed the remaining inconsistencies in the formatting through the whole document.

4. Please modify the location where the figures to be embedded as many figures are inserted in the middle of a paragraph.

Yes, we have modified the figures location and avoided embedding them in the middle of paragraphs.

Chai, F., Wang, Y., Xing, X., Yan, Y., Xue, H., Wells, M., Boss, E. (2021), A limited effect of sub-tropical typhoons on phytoplankton dynamics. Biogeosciences, 18(3), 849-859.

Wang, Y. (2020), Composite of typhoon induced sea surface temperature and chlorophyll-a responses in the South China Sea, Journal Geophysical Research: Oceans, 125, e2020JC016243.

**Reviewer 2**

In this work, Barrillon et al. characterize the response of phytoplankton to a storm event in the NW Mediterranean Sea. In my opinion, this work is truly relevant. There are few studies out there that compare the before and after of phytoplankton response to short-term anomalous events that disrupt the ecosystem. Since there is a real possibility that such extreme events may become more frequent in the future, there is a dire need for more studies on this topic. Unfortunately, most of these works occur as a reaction to a given extreme event and, thus, lack a comprehensive methodology that may evaluate the impact it had, which is understandable. This is not the case of this manuscript, as Barrillon et al. clearly tried to use as much as they could to characterize this event: in-situ ship-based sampling before and after the storm, a glider sampling during the storm, as well as remote sensing and modelled data to complement the in-situ data. Therefore, this is an important work and a good example on how various sources of data should be integrated to study a short-term event.

While methodology is sound, the writing is overall good and its conclusions are relevant and supported by the results, I do have a few gripes with the manuscript that I believe should be resolved before being accepted. Therefore, for my part, I recommend major revisions.

We are very grateful to the reviewer for the careful reading and relevant comments about the manuscript. All the remarks have been addressed below and the associated modifications are performed in the revised version of the manuscript.

(Color legend : comments and questions in green, answers in blue, new text in orange)

I will now list my main questions or areas where I think the manuscript could be improved.

• In the introduction, the goals of the work are not explicitly stated. There is a large paragraph detailing the FUMSECK cruise, some overall methodology and its aims, but these are the cruise's aims, not this work's aims. Clearly stating the objectives and linking them with the methodology and results would help the reader navigating through the substantial number of results described in this work.

Yes, the new introduction is now as follows, also taking into account the other comments about introduction (changes in orange).

"Marine environments are subject to short-term events whose effects on biogeochemical processes can be substantial. This is the case of desert dust deposit on oligotrophic areas (Guieu et al., 2014), volcanic ash deposit (Hammes et al., 2010), submarines sources of iron (Guieu et al., 2018), and sudden mixing of the water column from typhoons (Wang et al., 2020). Even the classical phytoplankton spring bloom can vary in intensity and spatial extent depending on the amount of previous short term storms (Ferreira et al., 2022). The effect of these processes on marine micro-organisms such as phytoplankton includes sudden changes in diversity and abundance. Depending on the redistribution of nutrients, the intensity of turbulence, the light conditions and the mixing of different water masses, the phytoplankton community can either collapse or grow, affecting carbon export by

generating decoupling phenomenon between production and remineralization (Henson et al., 2019).

Meteorological impulse wind events such as storms, and their effects on oceanic physics and on biogeochemistry, are poorly explored with in situ data. Such events generate mixing and stirring of the surface layer and can trigger transitional peaks in primary production, mainly explained by nitracline shoaling and grazers dilution (Lomas et al., 2009; Menkes et al., 2016). In oligotrophic ocean conditions, Babin et al. (2004) and Han et al. (2012) observed sudden and large increases in chlorophyll-a (chla) from satellite ocean color, lasting several weeks, after summer hurricane-storms. The resulting increase in chla integrated over the first few meters reached values close to those from the spring bloom (Babin et al., 2004) with potential primary production comparable to the one induced by some mesoscale (~10 - 100 km horizontal range) eddies. Nevertheless, the authors were limited in their interpretation by the lack of in situ observations. Only a few studies have combined high-resolution physical descriptions of wind events with a phytoplankton resolution at the functional group level. Some coastal studies, such as Fuchs et al. (2022), have evidenced pico-nanophytoplankton abundance and biomass responses, positive for most phytoplankton groups, within two to four days following wind-induced events at a coastal station located in the north-western (NW) Mediterranean Sea in stratified conditions. The authors showed that extreme events can generate daily biomass increases of the same order of magnitude as those observed during the spring bloom. Similarly, Anglès et al. (2015) studied the response of nano-microphytoplankton to tropical cyclones generating wind-physical forcing and substantial rains in the Western Gulf of Mexico. They highlighted strong increases in plankton abundance following the storms with delays consistent with Fuchs et al. (2022). These storms observed in either coastal Mediterranean systems or tropical open ocean may also exert a strong control on both primary production and community structure in the Mediterranean open ocean, thus playing a potentially important biogeochemical role on the whole basin. However, to our knowledge, no such event in the Mediterranean open ocean has been reported in the past.

The classical spring bloom as observed in temperate oceans is triggered by the shoaling of the mixed layer when passing from the winter convection to the spring stratification (Behrenfeld, 2010). The bloom ends when no more nutrients are available in the euphotic layer or when grazers overpass the phytoplankton growth capacity. This is particularly the case in the NW Mediterranean Sea characterised by winter deep convection (Houpert et al., 2016; Testor et al., 2018) and by spring blooms of different intensities that can be detected from satellite images (d'Ortenzio and Ribera d'Alcalà, 2009; Mayot et al., 2016). This area is affected by strong northerly winds, which strength in winter defines the bloom intensity (Conan et al., 2018). In summer stratified conditions, impulse wind events could induce submesoscale (~1 - 10 km horizontal range) vertical mixing and trigger patches of high phytoplankton production. Yet, observing the effect of these events on phytoplankton dynamics and distribution is challenging, especially during stratified oligotrophic conditions, and requires the deployment of dedicated automated and high-frequency sampling tools. Indeed, the mixing of the water column may bring microorganisms from deep to surface layers, affect their physiological properties due to photoacclimation processes, and impact their carbon-to-chlorophyll ratios used to run primary production models at large scales (Sathyendranath et al., 2020). In addition, some scarce observations at the functional group level evidence daily adaptation processes rather than community changes after water column mixing (Thompson et al., 2018) or a taxonomical dependency in physiological strategies (Graff and Behrenfeld, 2018). Being able to monitor phytoplankton distribution at a functional level, by integrating small and rapid scale dynamics into larger space and time scales would precise the role of phytoplankton in biogeochemical processes.

The objective of this paper is to study in situ physical and biological effects of a particularly intense episode of wind in spring 2019 in the Ligurian Sea (NW Mediterranean Sea) during the FUMSECK cruise (Facilities for Updating the Mediterranean Submesoscale - Ecosystem Coupling Knowledge, https://doi.org/10.17600/18001155, Barrillon et al. (2020)). High-resolution physical properties in the water column and surface phytoplankton functional groups distribution were combined to show abrupt changes in water characteristics, surface phytoplankton abundances and physiological indicators."

• For a work in which its conclusions revolve around the "role of storms on the biogeochemistry and ecology of the Mediterranean open sea (...)", I saw very few references in the introduction to works focusing on other than phytoplankton abundance or biomass changes. For instance, the authors could have discussed other studies that have approached the potential impact of such short-term events on carbon export (e.g., Hamme et al., 2010; Henson et al., 2012, 2013; Ferreira et al., 2022). Regarding this matter, I would be curious to see some remote sensing POC images/data before and after the storm. These may even tie in nicely with the conclusions of the article, if they reveal something interesting.

We thank you for the suggested references. After reading the suggested papers, we have added in the introduction several examples to illustrate the impact of short events on biogeochemical processes (first paragraph of the new introduction):

We also included a paragraph in the discussion:

"This could in turn foster the integrated primary production by enhancing phytoplankton division rate and biomass (Behrenfeld, 2010) which, when grazers are diluted, is related to higher organic carbon export efficiency (Henson, 2019). This phenomenon was also observed after winter storms in the Sargasso Sea, where diatoms increase was maximal within two days after shoaling of the mixed layer depth (Krause, 2009). These pulsed production events could be responsible for up to 20% of the global primary production in the Sargasso Sea (Lomas et al., 2011)."

Regarding the POC analysis, we found this NASA product : https://oceancolor.gsfc.nasa.gov/atbd/poc/#sec\_6, and analysed it.

Despite some cloud issues, we could find some increase in the POC just after the storm. This is now in the revised paper (last paragraph of the discussion):

"Our observations captured the short-term physical and phytoplankton response to a storm, with rapid and strong changes observed but without the possibility to follow in situ post-conditions. Although not representative of what happens in the entire mixed water column, satellite data showed an effect on surface temperature and chla within the ship-glider storm geographical zone (longitudes between 8° E and 8°30' E and latitudes between 43° 30' N and 43° 42' N). In this zone, the mean value of SST was lower during four days following the storm (6-10 May, 14.8° C) than between the 20 April-20 May period

(15.1° C), while the mean value of ChI\_ACRI was higher (0.44 ng mL-1 with respect to 0.32 ng mL-1), suggesting that the pico-nanophytoplankton size classes could have had the time to grow and accumulate, as their growth rate is close to one to two divisions a day when nutrients and light are available ((Morison et al., 2019). This is supported by the increase in Particulate Organic Carbon concentration (POC, https://oceancolor.gsfc.nasa.gov/atbd/poc/#sec\_6, MODIS Aqua L3 product) in the considered zone. Indeed, a higher value of POC concentration is observed during the four days following the storm (6-10 May, 104.1 ng mL-1) compared to the 20 April-20 May period (88.7 ng mL-1), suggesting that the whole trophic chain may be impacted by the storm."

• Some paragraphs or portions of the manuscript are a bit verbose and could be shortened or even removed. For instance, in lines 53-64, there is an exhaustive description of the FUMSECK cruise. This description could be shortened and most of it could be integrated into the Material and Methods section to avoid redundancy.

Yes, we have removed this paragraph from the introduction, shortened and embedded it in the material and methods introduction as follows :

"The FUMSECK cruise aimed at simultaneously sampling physical and biogeochemical data for the study of mesoscale and submesoscale dynamics, which imply structures such as eddies, filaments, or fronts over a horizontal spatial range of 1 to 100 km, a vertical one of 0.1 to 1 km, and a temporal range of days to a few weeks (Giordani et al., 2006; Ferrari and Wunsch, 2009; McWilliams, 2019). It took place from 30 April 2019 to 7 May 2019, in the Ligurian Sea (NW Mediterranean Sea), onboard the RV Téthys II. The circulation in the Ligurian Sea is generally cyclonic and characterised by a strong westward flowing geostrophic current along the coastline (Esposito and Manzella, 1982). The Northern Current(Millot, 1999), hereafter called NC, corresponds to the Northern branch of the current along the coastline. During this cruise, we deployed towed instruments (Moving Vessel Profiler, MVP) and an underwater glider (Testor et al., 2019) to measure physical properties at high resolution. These measurements have been paired with shipboard measurements of phytoplankton functional groups from an automated pulse-shape recording flow cytometer, based on cell sizes and pigment contents (Dugenne et al., 2014; Thyssen et al., 2014; Bonato et al., 2015; Louchart et al., 2020). Figure 1 shows the cruise and the glider trajectories, the MVP transects and the positions of the surface discrete sampling for nutrients and chla. A storm hit the region between 4 and 5 May. Right after the storm, during which we had to take shelter, the ship came back to the wind-exposed zone to collect data. Meanwhile, the glider stayed in the storm-exposed zone all along and collected data. In addition to in situ data, we exploited satellite data to guide the cruise and obtain a synoptic view of the region and a meteorological model to study the storm."

We also removed the mention to the stations, and the following paragraph :

"Several in situ instruments for measuring physics and biogeochemistry were deployed and are described in this section within the first two parts: transect measurements, and glider. The satellite data exploited to guide the cruise and obtain a synoptic view of the region are described in the third part, followed by the meteorological model. The last part deals with the comparison of the fluorescence and chla concentrations from the different measurements."

Also, passages such as 66-69 and 73-76 are redundant. There is no need to state what the results section will show after the material and methods because that will become clear for the readers as they continue to read the manuscript. I recommend looking at such situations across the manuscript to keep the text as straightforward as possible for the reader.

Yes, such passages are removed from the introduction, and the material and methods introduction.

• I think the sampling scheme could be clearer. For instance, there are underway surface water measurements of ADCP, SSS, SST and chl-a via fluorometer throughout the entire cruise (30/04/2019-07/05/2019). Then there is also an MVP which was deployed along seven different transects (only two are shown, as far as I noticed) and again sampled temperature, salinity, chl-a via fluorometer. It also included a plankton counter. Yet we only know the timing and duration of transect 1 (30/April) and 7 (5-6/May). Figure 6 is exhibiting the transects and its measured variables, but this should be clearly stated. Furthermore, while I was reading the manuscript, I was frequently unsure if what is being shown is the temperature/salinity data from the MVP or from the underway system. Inorganic nutrients were also sampled at 26 locations (or, at least, 26 samples were collected) and chl-a again (now via laboratory fluorometer) was sampled at 20 locations, all at surface. Judging from Figure 7, I think most of these chl-a and nutrients samples match, but, again, it should not be necessary for the reader to carefully compare figures and count stations to understanding the overall picture of what was done and how. What are the seven stations presented in Figure 1? Did multiple-variable sampling occur in these stations or are they just the location where the ship turned and began a new transect? For instance, in Figure 7, does the discrete in-situ sampling stations match the seven stations in Figure 1? Overall, I suggest revisiting the methodology section. One idea that may help could be including a table that lists all variables sampled, abbreviation and the source (ship underway, MVP, glider, discrete).

To answer your comment we did several actions for the revised manuscript:

- The stations correspond to vertical velocity measurements, which are not exploited in this paper  $\rightarrow$  we removed them from most of the figures and the text.
- We added two zoomed plots as Fig1b and Fig1c, one showing the MVP transects and the other one showing the in situ samplings locations (one color for the common Chl/nutrients stations, another color for the stations not in commun)
- We removed from the material and methods the measurements not used in the paper
- We added the table below at the end of Material and Methods, following your suggestions.

| Observable               | Abbreviation / name             | Vertical Range     | Sampling                                  | Source    |
|--------------------------|---------------------------------|--------------------|-------------------------------------------|-----------|
|                          | ADCP currents                   | $18-562\mathrm{m}$ | all cruise, 0.4 km resolution             | VM-ADCP   |
| Horizontal currents      | geostrophic currents            | first meters       | daily, 2 April to 3 July 2019             | Satellite |
|                          | Θ_tsg                           | 2 m                | all cruise, 0.2 km resolution             | TSG       |
| Conservative Temperature | Θ_mvp                           | $0-308\mathrm{m}$  | 7 transects, 1.3 km resolution            | MVP       |
|                          | ⊖_glider                        | $0-600\mathrm{m}$  | 2 transects, 1 km resolution              | Glider    |
|                          | S_tsg                           | 2m                 | all cruise, 0.2 km resolution             | TSG       |
| Absolute Salinity        | S_mvp                           | $0-308\mathrm{m}$  | 7 transects, 1.3 km resolution            | MVP       |
|                          | S_glider                        | $0-600\mathrm{m}$  | 2 transects, 1 km resolution              | Glider    |
|                          | RFluo_tsg                       | 2 m                | all cruise, 0.2 km resolution             | TSG       |
| Fluorescence             | RFluo_cyto                      | 2 m                | 400 samples, 3.9 km resolution            | AFCM      |
|                          | $\mathrm{FL}_{\mathrm{npq}}$    | $0-600\mathrm{m}$  | 2 transects, 1 km resolution              | Glider    |
|                          | Chl_tsg (converted)             | 2 m                | all cruise, 0.2 km resolution             | TSG       |
| Chlerenhall a            | Chl_insitu                      | 2 m                | 20 samples                                | in situ   |
| Chlorophyli-a            | Chl_cyto (converted)            | 2 m                | 400 samples, $3.9 \mathrm{km}$ resolution | AFCM      |
|                          | Chl_ACRI                        |                    |                                           |           |
|                          | Chl_MEDOCL3                     | first meters       | daily, 2 April to 3 July 2019             | Satellite |
|                          | Chl_MEDOCL4                     |                    |                                           |           |
|                          | Phosphate ( $PO_4^{3-}$ )       |                    |                                           | in situ   |
| Natalanta                | Nitrate ( $NO_3^-$ )            | 0                  | 26                                        |           |
| Nutrients                | Nitrite $(NO_2^-)$              | 2 m                | 26 samples                                |           |
|                          | Silicate (Si(OH) 4 ) |                    |                                           |           |
| Phytoplankton            | Abondance, size,                | 2 m                | 400 samples, 3.9 km resolution            | AFCM      |
|                          | biovolume, biomass              |                    |                                           |           |
| Wind intensity           | U 10                 | 10 m above surface |                                           |           |
| Heat Flux                | $Q_{\rm net}$                   | surface            | all cruise, nourly, 2 km resolution       | model     |

Table 1. Summary of the cruise measurements.

• Finally, I think the discussion can be improved since it seems slightly superficial. For a large body of results (pages 9-21, including figures), a ~1 page discussion is quite short, particularly when the results are good. I feel the discussion lacks a comparison to other works on storm events, both in the Mediterranean and other areas. The authors do briefly compare some results with the OSCAHR cruise, yet this cruise occurred in November and did not sample a storm event (as far as its mentioned in the manuscript). Therefore, why would the results be directly comparable? This is not to say that this comparison is not valuable, but a better contextualization should be included.

We agree that we had to deepen the discussion. We added a comparison with the datasets collected by Boudgriga et al., 2022, crossing the area during a similar period of the year (see the modified discussion at the end of the major comments below), and Latasa et al., 2022, showing similar trends in the Western Mediterranean Sea.

 Also, some conclusions within the discussion feel rushed and could do with better contextualization and arguments. For instance, in lines 317-318, the authors state that 'This suggests that cells did not have time to photo-acclimate or that different species were involved" after comparing the ratio of chl-a between chl-a in "cold" and "warm" waters. First, this information is not enough to make these statements. Secondly, this is the only mention of photoacclimation in the manuscript, except for line 46 in the introduction. Finally, this conclusion is quickly forgotten since the paragraph moves on and compares the increase in chl-a with a previous work from 2000.

We agree this part was missing information. A change is made in the new discussion.

 Again, in lines 333-334, the authors now suggest the drop in carbon/chl-a ratio is a "clear signature of a sudden change in phytoplankton cell physiology and translated the unadapted configuration of the cells to high light condition". Why is it a clear signature? Why is one thing related to another? It is up to the authors to make the 'bridge' between the results and the conclusions, not the reader. Moreover, the paragraph ends with this sentence, without any comparison to other studies or without a discussion of its implications.

In accordance with the previous comment, changes are made in the text in order to make the reading more friendly.

**New discussion:**

"In the NW Mediterranean Sea, in May, the water column is generally well stratified with nearly undetectable surface nutrient availability (Pasqueron De Fommervault et al., 2015). This was indeed the oceanographic setting before an intense storm dominated by north-westerly winds impacted the water column. The analysis of 30 years of coastal data in the South of France (Toulon and Marignane) by Meteo France shows that the typical periods of intense wind occur at the end of winter and middle of autumn. In Toulon, winds of intensity > 27.8 m s-1 occur on average 8 times per year, but only once every 4 years in May; winds of intensity > 36.1 m s-1 occur on average once every 2 years, and once every 30 years in May. The total occurrences of different wind intensities for the whole 1981-2010 period are shown in Table 3. The wind intensity of the studied storm, reaching a maximum of 36.1 m s-1, was rare in the Mediterranean Sea, and similar to the average wind intensity of the typhoons studied by Wang (2020).

| Location  | Wind intensity | $> 27.8{\rm ms^{-1}} \\ (> 100{\rm kmh^{-1}})$ | $> 30.6 \mathrm{ms^{-1}}$
(> 110 km h -1 ) | $> 33.3 \mathrm{ms^{-1}}$
(> 120 km h -1 ) | $> 36.1 \mathrm{ms^{-1}}$

[revised manuscript text omitted]

**Minor comments (lines on the left):**

3: Please remove or change 'violent' for a more adequate term (e.g., intense).

Yes, intense.

4: NW is written as 'north-western', yet the title includes 'northwestern'. Uniformize.

Yes, we uniformized all over the text.

8: missing of: factor of two

Yes.

9: missing of: and of seven

Yes.

24: missing have: have combined

Yes (see new introduction).

26: what does 'have evidenced pico-nanophytoplankton abundance and biomass responses' mean? Did it increase, lower?

Yes, an increase for most of the groups (see new introduction).

29: remove have: 'have studied'

Yes.

34: Are you suggesting that no previous cases of storms shaping primary production and phytoplankton community structure have been reported? It is not clear if this only refers to the NW Mediterranean, the entire Mediterranean or if it also includes other systems.

It refers to the Mediterranean open sea, changed in the new introduction.

38: missing the: overpass the phytoplankton growth capacity

Yes.

38: you already have north-western written in line 27, you can already use NW

Yes.

41: This area

Yes.

45: Add 'may': 'the mixing of the water column may bring microorganisms from deep to surface layers and affect their photophysiological properties (...)'

Yes.

62-64: methods?

Yes, this paragraph is now moved to Material and Methods.

91: were performed

Yes.

95 and 102: please specify that these are surface-only samples

Yes.

Figure 2 caption: Orgnano and Unidentified particles groups have the same colour (green dots). Use light and dark green, for instance, to differentiate them in the caption.

The Orgnano and Unidentified particles already had dark and light green colors, respectively, but as the Orgnano are rare and in the top-right part of the cytogram (Fig.2a), they are not visible enough. We changed the color of Orgnano to pink.

149-150: It should have been calibrated prior to the cruise. Nevertheless, how good is the agreement with ship-based chl-a? Since the glider is the only source of data during the storm, this should be presented as supplementary material or, at least, the R, p-val, error and N should be indicated in the text.

We put the following paragraph and figure in the revised paper, Appendix A.

"No pre-cruise calibration of the Ecopuck was carried out. Nevertheless, we observe a good statistical agreement between the measurements of the glider with those taken from the ship (Fig. A1). Over a sample of N = 33 glider profiles where the glider-ship distance is lower than 40km, surface chla from the ship TSG (Chl\_tsg) and chla 0-10 m average from the glider have a correlation coefficient of R = 0.76 (with a significant p-value of 2.5e-7) and a mean standard error of 0.067 ng mL-1, well below of the amplitude of the observed signal during the storm (maximum Chl\_tsg of 1.11 ng mL-1).

Values from the onset of the storm have been excluded (grey values after 5 May) since the glider was experiencing different conditions than the ship sheltering from the bad weather. At the end of the time series, when the glider was recovered, the values between the two platforms agree well again, which gives us a good confidence in the chla fluorescence signals described by the glider's sensor during its mission."

Figure A1. Comparison of chla between the ship TSG and the 0-10 m average from the glider. The numbers indicated on top of the figure and the blue markers correspond to measurements where the glider-ship distance is lower than 40 km.

156: swap SSH and sea surface height

Yes.

157: swap SST and sea surface temperature

Yes.

157: there is no such thing as sea surface chl-a. Satellite chl-a does not capture only surface chl-a.

Right, we have replaced sea surface chla by chla integrated over the first few meters, when satellite chla is concerned.

159-160: please provide a bit more detail on the satellite products instead of just referring to another paper. You may leave the citation, but please add a brief description, just mentioning the name of the products or sensors and their resolution.

Yes, you will find below the information on all the products, now in the paper in Appendix B.

- SSH and associated geostrophic currents

. "Mediterranean ocean gridded L4 Sea Surface Heights and derived variables" (SEALEVEL\_MED\_PHY\_L4\_NRT\_OBSERVATIONS\_008\_050, now SEALEVEL\_EUR\_PHY\_L4\_NRT\_OBSERVATIONS\_008\_060, https://resources.marine.copernicus.eu/product-detail/SEALEVEL\_EUR\_PHY\_L4\_NRT\_OBSERVATIONS\_008\_060/INE ORMATION): 0.125° x 0.125°, multi-satellite

**- SST**

- . "Mediterranean Sea high resolution and ultra high resolution Sea Surface Temperature analysis" (SST\_MED\_SST\_L4\_NRT\_OBSERVATIONS\_010\_004, https://resources.marine.copernicus.eu/product-detail/SST\_MED\_SST\_L4\_NRT\_OBSERVATIONS\_010\_004/INFORMAT ION): 0.01° × 0.01°, nighttime images, multi-satellite

**- Chla**

- . "Global ocean Chlorophyll from satellite observations" (OCEANCOLOUR\_GLO\_CHL\_L3\_NRT\_OBSERVATIONS\_009\_032, now OCEANCOLOUR\_GLO\_BGC\_L3\_NRT\_009\_101,
- https://resources.marine.copernicus.eu/product-detail/OCEANCOLOUR GLO BGC L3 NRT 009 101/INFORMATION): 4km x 4km, ACRI-ST company, multi-satellite, hereafter called ChI\_ACRI
- . "Mediterranean Sea surface Chlorophyll concentration from multi satellite observations" (OCEANCOLOUR\_MED\_CHL\_L3\_NRT\_OBSERVATIONS\_009\_040, now OCEANCOLOUR\_MED\_BGC\_L3\_NRT\_009\_141,
  - https://resources.marine.copernicus.eu/product-detail/OCEANCOLOUR\_MED\_BGC\_L3\_NRT\_009\_141/INFORMATION) : 1km x 1km, multi-satellite, hereafter called Chl\_MEDOCL3
- . "Mediterranean Sea daily interpolated surface Chlorophyll concentration from multi satellite observations"
  - (OCEANCOLOUR\_MED\_CHL\_L4\_NRT\_OBSERVATIONS\_009\_041, now OCEANCOLOUR\_MED\_BGC\_L4\_NRT\_009\_142,
  - https://resources.marine.copernicus.eu/product-detail/OCEANCOLOUR\_MED\_BGC\_L4\_NRT\_009\_142/INFORMATION) : 1km x 1km, multi-satellite, hereafter called ChI\_MEDOCL4

**167: reference for the ECMWF model?**

We added these two references :

- Bechtold, P., R. Forbes, I. Sandu, S. Lang, and M. Ahlgrimm, 2020: A major moist physics upgrade for the IFS. ECMWF Newsletter, No. 164, ECMWF, Reading, United Kingdom, 24–32, https://www.ecmwf.int/node/19720.

- Ben Bouallègue, Z., 2020: Accounting for representativeness in the verification of ensemble forecasts. ECMWF Tech. Memo. 865, ECMWF, 28 pp., https://www.ecmwf.int/node/19544.

**174: techniques instead of sources**

**Yes.**

179: in this context, this R2 could be higher.

We think we had misqualified the names in the equation. We have changed the Chl\_tsg and Chl\_cyto by the Chl\_insitu.

→ "Fluorescence from the TSG (RFluo\_tsg) was converted into units of chla concentration (Chl\_tsg, ng mL-1) using the significant correlation with Chl\_insitu, Chl\_insitu = 0.85 x RFuo\_tsg - 0,19, r2 = 0.79, n = 20. AFCM chla concentration (Chl\_cyto) was estimated from the Rfluo\_cyto. Values were normalised with 2 µm Polyscience beads, and multiplied by the abundance of each group to get the total normalised Rfluo\_cyto per unit of volume (nRFluo\_cyto (a.u mL-1)). nRFluo\_cyto was then compared to the Chl\_insitu (Fig. 3a and b). A set of samples from a minicosm experiment (PIANO, unpublished data), acquired with the same chla extraction protocol and the same Cytosense instrument was added to the observations. These samples presented higher chla concentration values, strengthening the relationship. The linear relation between nRfluo\_cyto and Chl\_insitu was used to estimate chla concentration for each AFCM phytoplankton group (Chl\_cyto, ng mL-1) following the linear regression Chl\_insitu = 0.11 x nRFluo\_Cyto, r2 = 0.97, n = 41 (Fig. 3b). The origin of the linear regression was not significantly different from zero."

In the case this was not the reviewer's request, we found the correlation between chla from samples analysed in the lab and fluorescence from a fluorimeter not so bad in our case. Indeed, as a comparison:

Marrec et al., 2016 :  $r^2$ =0.50, n=41, chla varying from 0.08 to 0.41 µg L-1

Thyssen et al., 2015 :  $r^2$ =0.86, n=12, chla varying from 0.21 to 7.80 µg L-1

Our study :  $r^2 = 0.79$ , n=20, chla varying from 0.07 and 0.82 µg L-1

189: again, remove sea surface.

Yes.

191-192: the comparison period should actually be much shorter since the main ocean colour sensors overpass occurs between 10h-13:30h, depending on the sensor (see section 3.1 in Sathyendranath et al., 2019; Remote Sensing, 19(19), 4285). I would try rerunning the comparisons with a shorter period, it is possible the results may improve.

We agree that the 10h-13h30 period is indeed more appropriate to perform the comparison: most of the correlations found with this time period increase with respect to the 6am-6pm time period. Yet, using this time period, the correlation between

Chl\_insitu and Chl\_ACRI is performed on 4 points only, between Chl\_insitu and MEDOC\_L3 on 2 points and between Chl\_insitu and Chl\_MEDOCL4 on 5 points, which is far too low for this time period to be used in practice.

Figure 3:

 how does the R between MEDOCL3 and MEDOCL4 is equal to 1, but the R between MEDOCL3 and Chl\_insitu is 0.84 and MEDOCL4 and Chl\_insitu is 0.65?

MEDOCL3 has gaps due to cloud coverage, and MEDOCL4 fills the gap with some climatology. Thus, L4 points are composed of the L3 points plus some additional climatology-based interpolated points. The correlation between L3 and L4 is therefore performed only on the "L3 points": by construction the correlation is 1. Similarly, the Chl\_insitu/MEDOC\_L3 correlation is performed on 10 points whereas the Chl\_insitu/MEDOC\_L4 correlation is performed on 17 points (10 "L3 points" and 7 interpolated points). The worst correlation for Chl\_insitu/MEDOC\_L4 shows that the climatology interpolation does not match our in situ observations here.

• Where does the N=4555 come from when comparing satellite and in-situ data? Satellite data should be, at most, daily data unless the authors are working with geostationary sensors

We agree that this number can be misleading. Satellite data are indeed daily provided, on a lat/lon grid. We performed the association between each Chl\_tsg data and the Chl satellite data on the same day and for the closest lat/lon pixel, then selected the ones where the Chl\_tsg data is between 6:00-18:00 UTC day time. As a matter of fact, checking this, we found that the number 4555 in the text was not correct: Chl\_ACRI n = 3522, Chl\_MEDOCL3 n = 2094, Chl\_MEDOCL4 n = 4498. We corrected these numbers.

We also added this sentence in the corresponding paragraph :

"We performed the association between each ChI\_tsg data and the corresponding ChI satellite data on the same day and for the closest lat/lon pixel, then selected the ones where the ChI\_tsg data are between 6:00-18:00 UTC daytime, to minimise the effect of night extrapolated points."

• The colour palette for the correlation plot should be changed to a more uniform one (e.g., R=0 white, R=1 dark red)

Yes, we changed to shades of blue.

197-199: these are not results

Yes, we moved these lines to the Material and Methods.

201: why did you opt for MEDOCL4 when the relationship between satellite and in-situ was much better for MEDOCL3?

For this figure, the purpose was to define the mean dynamic zones during the cruise. We decided to use MEDOCL4 even if the correlation with in situ Chl is worse than the MEDOCL3 one, to avoid the clouds that can create artificial features when averaging on several days.

204: I recommend changing the Chl-a units from ng/mL to either ug/L or mg/m3 since these options are more commonly used.

We understand your remark, as chlorophyll-a concentration is often written in  $\mu$ g/L. Our manuscript uses volumes a lot, and we are presenting all the data in units/mL to homogenise with the flow cytometry datasets. Indeed, if we would use L, we would be required to add a 103 for each abundances presented in the tables and in the figures.

212: I recommend using m/s for wind speed. Also, the same units should always be used throughout the text (see line 222).

Yes, we used m/s and homogenised through the text.

212: are these average or maximum intensities? Not clear.

They are the ranges of the intensities.

215-218: Again, these are not results from this work, unless you include them as supplementary material. Thus, this comparison would be more suitable in the discussion.

Yes (see new discussion).

224: The final sentence of the paragraph can be removed.

Yes.

237: rose instead of rised up

Yes.

299-300: this should also be in the discussion.

Yes.

302: the water column was

We mean general characteristics, we rephrased :

"At the time of the FUMSECK cruise, in May, the water column is generally well stratified with nearly undetectable surface nutrient availability (Pasqueron De Fommervault et al., 2015"

 $\rightarrow$  "In the NW Mediterranean Sea, in May, the water column is generally well stratified with nearly undetectable surface nutrient availability (Pasqueron De Fommervault et al., 2015, This was indeed the oceanographic setting before an intense storm dominated by north-westerly winds impacted the water column."

311-312: add percentages or values when comparing

Yes (see new discussion).

---

## Author Response (AR2)

EDITOR

Dear authors,
Thank you for submitting the revised version of your manuscript. Both reviewers recommend publication but referee 2 provides a list of edits that should be taken into account when preparing the final version. I agree with their comment that some figures should be moved to the appendix (for instance, Figs. 2 and 3). Also, the quality and resolution of all figures should be checked thoroughly.
Best regards,
Emilio Marañón

Dear Editor,
The authors would like to thank you and the referees for your time and work on this paper. All the comments have been carefully taken into account. We agreed to put Fig 2. on appendix, but we would rather keep Fig. 3 and Fig. 14 as they are. Fig 3. shows that all our measurements and methods for chla agree, which we think is important for supporting the results. We would prefer to keep it in the core of the paper. Fig. 14 is showing the sudden change in the photo physiology and is a support to a large part of the discussion, we would also prefer to keep it as it is.
Best regards,
Stéphanie Barrillon.

**Anonymous during peer-review: Yes** No

**Anonymous in acknowledgements of published article: Yes** No

Recommendation to the editor

| | |
|---|---|
| **1) Scientific significance**
Does the manuscript represent a substantial contribution to scientific progress within the scope of this journal (substantial new concepts, ideas, methods, or data)? | Excellent **Good** Fair Poor |
| **2) Scientific quality**
Are the scientific approach and applied methods valid? Are the results discussed in an appropriate and balanced way (consideration of related work, including appropriate references)? | **Excellent** Good Fair Poor |
| **3) Presentation quality**
Are the scientific results and conclusions presented in a clear, concise, and well structured way (number and quality of figures/tables, appropriate use of English language)? | Excellent **Good** Fair Poor |

For final publication, the manuscript should be

**accepted as is**

accepted subject to **technical corrections**

accepted subject to **minor revisions**

reconsidered after **major revisions**

**rejected**

**Were a revised manuscript to be sent for another round of reviews:**

**I would be willing to review the revised manuscript.**

I would not be willing to review the revised manuscript.

**Suggestions for revision or reasons for rejection (will be published if the paper is accepted for final publication)**

The authors have substantially improved the manuscript and I recommend the paper for publication.

The authors are deeply thankful for your expertise and time on this paper.

**Anonymous during peer-review: Yes** No

**Anonymous in acknowledgements of published article: Yes** No

**Recommendation to the editor**

| | |
|---|---|
| **1) Scientific significance**
Does the manuscript represent a substantial contribution to scientific progress within the scope of this journal (substantial new concepts, ideas, methods, or data)? | **Excellent** Good Fair Poor |
| **2) Scientific quality**
Are the scientific approach and applied methods valid? Are the results discussed in an appropriate and balanced way (consideration of related work, including appropriate references)? | **Excellent** Good Fair Poor |
| **3) Presentation quality**
Are the scientific results and conclusions presented in a clear, concise, and well structured way (number and quality of figures/tables, appropriate use of English language)? | Excellent **Good** Fair Poor |

For final publication, the manuscript should be

accepted as is

accepted subject to technical corrections

**accepted subject to minor revisions**

reconsidered after major revisions

rejected

**Were a revised manuscript to be sent for another round of reviews:**

I would be willing to review the revised manuscript.

**I would not be willing to review the revised manuscript.**

**Suggestions for revision or reasons for rejection (will be published if the paper is accepted for final publication)**

I am thankful to the authors for taking the time to thoroughly revise the manuscript. Having reread the article, I feel that the manuscript has clearly improved and that the authors have included new relevant information both in the text and the supplementary material. As such, I have only very minor suggestions.

The authors are deeply grateful for your reading and comments. Answers are inlined in your suggestions.

Minor comments:

If possible, make an effort to slightly reduce the number of figures in the main text. I'm sure some of the more technical figures can be added to the Supplementary Material. Example, Fig 3 and 14.
We decided to put Fig. 2 in appendix, but we would rather keep Fig. 3 and Fig. 14 as they are. Fig. 3 shows that all our measurements and methods for chla agree, which we think is important for supporting the results. We would prefer to keep it in the core of the paper. Fig. 14 is showing the sudden change in the photo physiology and is a support to a large part of the discussion, we would also prefer to keep it as it is.

Line 23: keep only turbulence (no need for "intensity of")
Yes

Line 24: remove either
Yes

Line 25: phenomena
Yes

Line 26: add comma before "such as storms"
Yes

Line 42: add space between 'may' and 'also'
Yes

Line 66: the font size of the doi link does not match the remainder of the text
Yes, we checked and corrected for all the website links.

Line 164: I suggest just leaving 'chla concentration", removing 'integrated over the first few meters', since the integration actually depends on the first optical depth which can be over a few tens of meters in clear waters.
Yes

Line 166: again, different font size
Yes

Line 197: remove 'integrated over the first few meters'
Yes

Line 235: here you have a space between the value and the units. However, in other instances, there is not a space. Please uniformize along the text.
We uniformised and checked all along the text.

Table 1: caption should be on top of the table. Also, please add a bit more information on caption (e.g. summary of the variables measured during the cruise, including their sources, their sampling spatial and temporal resolution, and the vertical range along which they were measured).

Yes, we also put all the table captions on top.

Figure 11: It might be just a matter of the pdf compression, but the resolution of this figure seems low. This is particularly visible in the legends within each panel. Please, check. Also, replace "tick labels" for "ticks' labels"

Yes, we replaced the figure with a better resolved version.

Line 320: Synechococcus should be in italic

Yes

Line 337: add coma before while

Yes

Line 401: modelling

Yes